# Reliability of a convolutional neural network in segmenting multiple sclerosis lesions from MRI: Impact of data augmentation, image modality and tolerance with U-Net architecture

Adam C. Szekely-Kohn[1,2]*, Marco Castellani[1], Luca Baronti[1,2], Zubair Ahmed[3,4], William G. K. Manifold[5], Michael Douglas[3,6,7], Daniel M. Espino[1]

1 School of Engineering, University of Birmingham, Edgbaston, Birmingham, United Kingdom, 2 School of Computer Science, University of Birmingham, Edgbaston, Birmingham, United Kingdom, 3 Institute of Inflammation and Ageing, University of Birmingham, Edgbaston, Birmingham, United Kingdom, 4 University Hospitals Birmingham NHS Foundation Trust, Edgbaston, Birmingham, United Kingdom, 5 Royal North Shore Hospital, St Leonards, Sydney, New South Wales, Australia, 6 School of Neurology, Dudley Group NHS Foundation Trust, Russells Hall Hospital, Birmingham, United Kingdom, 7 School of Life and Health Sciences, Aston University, Birmingham, United Kingdom

* axs1090@student.bham.ac.uk

## Abstract

Multiple Sclerosis (MS) is an inflammatory demyelinating disease of the central nervous system, typically exhibiting radiologically identifiable lesions within the brain and spinal cord, features key to both diagnosis and clinical disease monitoring. Manually identifying and segmenting lesions is both difficult and time consuming, thus optimising approaches to reliably automate segmentation is highly beneficial. The aim of this study was to assess the impact of data augmentation and manipulation on the accuracy of automated lesion segmentation using MRI scans from MS patients. Factors examined include MRI modalities in both isolation and combination, image augmentation, lesion size and size of testing set relative to training. The MICCAI 2016 MS dataset was used in this study, with U-Net chosen as the algorithmic method for segmentation. Each factor was optimised and then combined to maximise segmentation accuracy; the Dice metric was used as the focal metric to assess the efficacy of any given permutation of the setup. Statistical significance was assessed using the Mann–Whitney U-test, with each permutation repeated five to ten times to ensure robustness. The best Dice score achieved using the testing and training dataset as outlined in the MICCAI 2016 challenge rubric was 0.59, approximately a 2% improvement against controls. To achieve this result whilst adhering to the training and testing distribution as defined in the dataset publication, the optimal imaging sequence was determined to be proton density. The augmentation conditions used included implementing an additional rotation of the dataset (doubling it in size) and excluding lesions $< 36.43$ mm$^3$ in volume. The impact of data manipulation and augmentation was found to be statistically significant against controls using a Mann-Whitney U-test

**Data availability statement:** Data used can be found in publicly accessible data repositories: (Shanoir), https://shanoir.irisa.fr/shanoir-ng/welcome The data used is already published under the name of 'MS lesions segmentation challenge of MICCAI 2016'. Therefore ethical approval was not required for this study. The code used can be found: doi.org/10.5281/zenodo.16211828.

**Funding:** This work was supported by the Engineering and Physical Sciences Research Council (EP/T517926/1 to ACSK). The funders had no role in study design, data collection and analysis, decision to publish, or preparation of the manuscript.

**Competing interests:** The authors have declared that no competing interests exist.

for lesion segmentation. An ancillary finding of this study was that there was no statistically significant difference between using one MRI modality for training and another for testing.

## Author summary

Multiple sclerosis (MS) is a disease of the brain and spinal cord with both inflammatory and neurodegenerative features impairing mobility, cognition and sensory function. Damage caused by MS is apparent radiologically in magnetic resonance imaging (MRI) scans, which are used by clinicians to diagnose and monitor the disease. Manually identifying these lesions in MRI scans is time-consuming and can vary between clinicians, so automated methods using machine learning are increasingly being developed to assist with this task. However, an often overlooked factor is how the MRI data are prepared before training these models. In this study, we examine how different preprocessing choices affect the performance of automated MS lesion segmentation. Using a diverse and well-known dataset, we investigate the impact of MRI modality, lesion size, and data augmentation on segmentation accuracy. We find that including proton density MRI scans, removing the smallest lesions, and adding rotated training images improves performance by around 2%. We also observe that models trained on one MRI modality can perform similarly when tested on another. These findings highlight how preprocessing choices can affect the reliability of automated tools and the potential for models to generalise across different MRI data.

## 1. Introduction

Multiple Sclerosis (MS) affects 2.8 million people around the world and is a major cause of physical disability in working age adults [1]. Despite major advances in the diagnosis and treatment of the condition, MS remains incurable [2], and is characterised by a mix of inflammatory damage and progressive neurodegeneration of the central nervous system (CNS) [3]. One characteristic manifestation includes the presence of inflammatory demyelinating foci (lesions) found in CNS tissues [4], which can be identified by means of magnetic resonance imaging (MRI) scans [5–7]. The heterogeneity (shape, size, number, location) of the lesions [8] make it challenging and time consuming to reliably identify and segment, for experienced clinicians [9,10]. This process is of key importance, when monitoring for disease activity and progression, with a significant impact on treatment pathways. Segmentation following imaging allows for the degree of demyelination to be assessed and automating this process may be beneficial to clinicians, saving time and potentially improving diagnostic precision [11].

Object identification in medical imaging has become a key challenge in modern radiology. In a clinical setting, critical judgements relating to MS diagnosis and

prognosis are still made for the most part by medical professionals, reviewing imaging series directly. Heterogeneous and irregularly shaped abnormalities like MS lesions are challenging to identify. Technological advancements, specifically in artificial intelligence (AI) have led to methods by which segmentation of regions of interest (ROI), such as demyelinating lesions (or other key anatomical or pathological structures such as tumours), can be automated [11]. Deep learning techniques, such as convolutional neural networks (CNNs) have proved to be highly effective in discovering hierarchical feature representation [12]. For the segmentation of MS lesions, techniques such as graph-cuts [13], multi-layer perceptrons [14] and K-nearest neighbours [15] have been used effectively. U-Net and U-Net inspired methods [16] have been applied successfully to a variety of medical objects of interest including organs like the liver, lungs, pancreas, bones, cancerous and pre-cancerous tissues [17–22] alongside central nervous system tissues and MS lesions within [23].

There are published studies on MS lesion segmentation involving U-Net and the Medical Image Computing and Computer Assisted Interventions (MICCAI) dataset [24–26]. All of them achieved a better result than those which accompanied the challenge publication [27]; however, these retrospective entries did not strictly adhere to the stipulations outlined in the original challenge. For instance, the majority of the data provided was unused, MRI volumes were deconstructed into their constituent slices, mixed, and training and testing datasets redefined on a random slice by slice basis. Only 15 of the 53 MRI volumes included in the challenge were used and these were divided between training and testing [24–26] using data from only three of the four different scanners from different scanners used to collect the MICCAI dataset. In all three cases, the individual 2665 slices comprising the fifteen MRI volumes, were split using a ratio of 1:4 for the testing and training, respectively.. Hence, the trained algorithms from their studies were less likely to be generalisable, and possibly gain advantages regarding evaluation metrics which are unlikely to be reproduced in a true clinical setting. Critically, this point highlights the benefit of benchmarking the input parameters to a neural network such as U-Net.

Two further studies have been carried out on MS lesion identification using U-Net but with the use of alternative datasets. These datasets are not publicly available, meaning that without access, results cannot be reproduced and cannot be used for comparison with studies using different algorithm architectures [28]. One such study used transfer learning, in which training was carried out using generic data to subsequently perform task-specific identification, to successfully segment lesions in MRI scans of MS patients. No information was provided to the network about the hyperintense lesions on which it was tested [23]. The results were noteworthy as CNNs were thought to be naive, yet still managed to perform the task to a high standard. The most relevant study analysed the impact of lesion size on accurate detection. It was found that a Dice score of 0.8 could be achieved if lesions $< 500 \mu$l were excluded in the detection process as they were considered to be too small for reliable detection automated or otherwise [28]. The effect of dataset size on the accuracy of detection was assessed using a large privately held dataset of 1008 patients scanned in 68 different centres [28].

The limiting factors to segmentation performance can be placed into one of two broad categories: hyperparameter or image related. Hyperparameters describe the complete architecture of a neural network not learned from training data and defined before the training process is initiated. They include elements as basic as the number of layers and neurons comprising a neural network as well as loss function, activation function and learning rate. A loss function quantifies the difference between a neural network output and its associated ground truth (in the case of medical segmentation - manual delineation performed by clinicians), essentially assessing performance and being used as an optimisation guide. Activation functions introduce non-linearity between layers, which is an integral property of a neural network for learning and discriminating data patterns. The learning rate determines the speed at which the neural network is trained and is responsible for minimising the loss function. In an ideal scenario the loss function should reach 0 indicating no difference between the neural network predictions and clinical opinion. Setting the learning rate is a balancing act. If it is too small, the learning algorithm can take an excessive amount of time to minimise the loss function. If it is too large it may introduce oscillations in the learning process and prevent the fine tuning response of the neural network. Each of these elements are critical design considerations that impact the capacity of a neural network to learn and its overall performance.

Data-related issues that affect algorithmic performance are also numerous and include: class imbalance, data resolution, ground truth accuracy and number of images or volumes. In the context of MS brain MRI scans, it is likely MS lesions constitute only a small region of the total scan, which is a difficult limitation to address. Improving ground truth requires the services of experienced clinical professionals. There are, however, simple methods by which smaller datasets can be enlarged, potentially permitting for automatic segmentation to be improved. Whilst there has been substantial research into the architecture and methods used for segmentation [27,29], there has been less emphasis on analysing how alterations to data can improve results in the context of MS lesion segmentation.

Basic rotational augmentations have been regularly used in the context of medical imaging to increase dataset size [24,30], exploiting the CNN property of not being rotationally invariant. One area that has yet to be investigated is the impact of using multiple MRI modalities simultaneously for training. A benefit of using the MICCAI 2016 dataset is that it contains five MRI modalities for each object [31]. An analysis of the differential effect of image type on segmentation performance has not been conducted and does not appear to exist in the literature. This provides an opportunity to utilise multiple image modalities to boost data quality and quantity.

The aim of this study was to evaluate the performance of U-Net in MS lesion identification and segmentation from MRI scans. Objective analysis focused on the Dice metric [32] as altered by image modality, data augmentation and exclusion criteria for MS lesion size. Firstly, FLAIR, T1, T2, gadolinium-enhancing contrast medium and proton density image modalities were evaluated individually and simultaneously, and assessed how training and testing on different image modalities impact segmentation performance. Secondly, expanding of datasets was evaluated artificially via rotational augmentations, and non-artificially by increasing the training data size (increased ratio of training to testing data). Finally, size-based exclusion of lesion features was evaluated, to assess the extent to which the minimum tolerance for segmentation impacts on performance metrics. While this study focuses specifically on MS lesions, a wider purpose of this study is to aid discussion between scientists, engineers and clinical practitioners, who are considering clinically applied computational techniques. This study, for instance, outlines benefits and limitations of computational techniques in circumstances of high variability. The dataset used includes seven different individuals undertaking manual segmentation and multiple scanners with different specifications used to collect data [27]. The current study adheres to the rubric outlined in the challenge [27] to prevent obfuscation of the efficacy of the method and the augmentations used. It also ensure that the method is directly comparable to other studies carried out using the same dataset; however, parts of the experimental design have been set-up which do deviate from the rubric to determine how doing so can skew the subsequent metrics.

## 2. Methods

The following section provides an overview of the dataset used in this study, the architecture of the neural network used and the procedural process in carrying out the computational experimentation (Fig 1).

Fig 1 illustrates the work procedure for this study. Initially literature was assessed on the application of machine learning to MS MRI on which a comprehensive review was published [11]. Identification and segmentation of lesions was identified as an area of application. A number of datasets were located [29,33,34] and the MICCAI 2016 dataset [31] was chosen as it was both sufficiently large and a substantial number of studies utilising it had already been published, meaning multiple existing points of comparisons. The U-Net model was chosen and a design of experiment performed [16]. Limited information on the the impact of data manipulation, pre-processing and MRI modalities was found and so it was decided to centre the experimentation on this literature gap. A rigorous statistical analysis was performed on each experiment with each permutation being run between 5 and 10 times allowing for use of Mann-Whitney U-tests to evaluate statistical significance.

### 2.1. Ethics statement

The authors of this study played no role in the collection of the dataset used. Since this was the case, and the dataset had already been made publicly available, ethical approval was not required. The data used in this study were generated

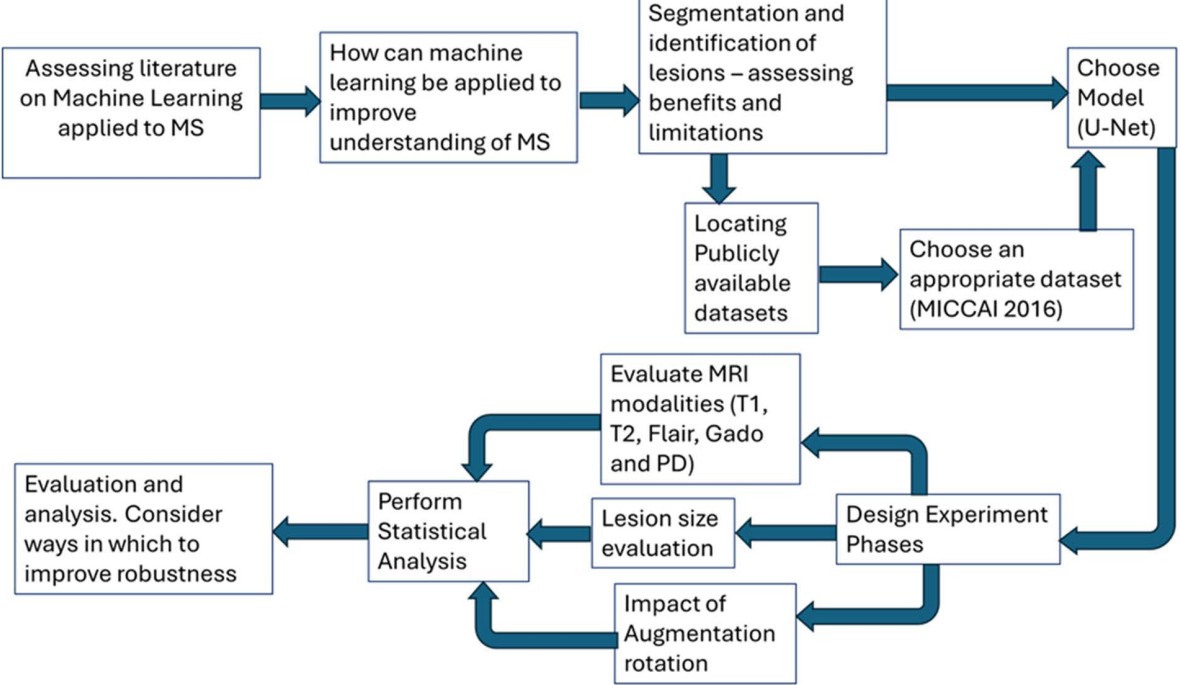

**Fig 1. A flowchart illustrating the pipeline for this study.**

by neurologists as part of the French MS registry (Observatoire Français de la Sclérose en Plaques, OFSEP). During the collection process, patients gave informed consent to OFSEP. Nominative information which could identify individuals was deleted from the MRI data prior to transfer and storage on the Shanoir platform (Sharing NeurOImaging Resources) [35]. Data were made available according to a specific license to ensure adherence to the European General Data Protection Regulation (GDPR). The main points outlined in the paper associated with the dataset [31] are that it is necessary to acknowledge the OFSEP publication [27]. Those who download the dataset agree to provide an email address so that OFSEP can keep track of its usage. Finally, users of the data agree not to redistribute it or use it more than three years after having downloaded it. The authors of this study have complied with all of the above. No data, besides that located in the MICCAI 2016 repository on the Shanoir platform, have been used.

## 2.2. Dataset

**2.2.1. Image acquisition.** The dataset used were made available through the 2016 MICCAI MRI MS Lesion Segmentation Challenge and comprised data from 53 MS patients [27,31]. The image volumes were of voxel dimensions 512× 512×144. The dataset included MRI scans of participants taken from hospitals across France, using a variety of MRI scanners (Table 1) [31]. For the total number of participants across all centres, the overall gender ratio was 38:15 (women to men). The parameters used to generate the MRI volumes, namely, time to echo (TE): the time taken between signal emission and detection of echo, repetition time (TR): the time between consecutive pulses and resolutions varied depending on the centre and scan type [31]. Patients were selected to represent a broad spectrum of lesion load and volume, enabling comprehensive capture of the varied consequences of demyelination [31].

**2.2.2. Imaging protocols.** Five types of MRI were collected for each patient: FLAIR, T1, T2, proton density - T2 weighted (referred to as proton density or PD images from here on in) and T1 post-gadolinium injection (referred to

**Table 1. MRI scanners and participant dataset information available from the MICCAI 2016 dataset [31].**

| Centre | Scanning Center Location | Brand | Magnet Strength | Training Cases | Test Cases | Gender Ratio W:M |
|--------|--------------------------|-------|-----------------|----------------|------------|-------------------|
| 01 | University Hospital, Rennes | Siemens Verio | 3T | 5 | 10 | 2.75:1 |
| 03 | University Hospital, Bordeaux | General Electrics Discovery | 3T | 0 | 8 | 7:1 |
| 07 | University Hospital, Lyon | Siemens Aera | 1.5T | 5 | 10 | 4:1 |
| 08 | University Hospital, Lyon | Philips Ingenia | 3T | 5 | 10 | 1.14:1 |

as gadolinium-enhanced from here on in). Examples of each of the aforementioned imaging modalities along with a respective ground truth indicating lesion location are depicted in Fig 2. Data was generated in the participating neurology centres using the framework of the Observatoire Français de la Sclérose en Plaques (OFSEP), the French MS registry [36]. A care protocol was used for MRI collection and informed consent from the patients was provided to OFSEP. A national French protocal for harmonisation of images was used, representative of the standards at the time of acquisition [37,38].

**2.2.3. Imaging procedures.** The ground truth delineations for the dataset was created by recruiting seven junior radiologists who carried out segmentation manually using the T2 and FLAIR images. A single consensus mask was then produced using software that combined delineation performance of each of the radiologists [31]. Test data was collected across four scanning centres from four different scanner whereas training data was collected from the three of the four centres (Table 1). A rubric was set the training to testing data ratio at 38:15. Any data which could potentially identify a study participant was deleted by those who collected the MRI scans before its upload to the aforementioned platform. According to the license, users may not upload the dataset files to another public online platform [31]. Further details on image acquisition, imaging protocols and imagine procedures can be found in the publication associated with the collection of the dataset [31].

## 2.3. Architecture

U-Net was created by Ronneberger et al [16]. The architecture of U-Net, when diagrammatically depicted, normally appears in the shape of a 'U' (Fig 3).

The U-Net architecture was taken from the Monai library. The 3D version of the neural network was used since the data type used were 3D MRI volumes. A single input channel was used. Five downsampling blocks were used containing two convolutions each connected by rectified linear unit (ReLU) activation functions. The blocks themselves were connected by max pooling layers. The kernel sizes for the convolution layers and pooling layers were 3×3 and 2×2, respectively. Four upsampling blocks were used, mirroring the downsampling half the neural network, ensuring that the output dimensions were the same as those of the input. The 'amsgrad' variant of the Adam optimizer with a learning rate of $10^{-5}$ were used. For further detailed procedure on U-Net see S1 Text.

## 2.4. Experimental procedure

The used algorithm was a basic U-Net. Four phases of experimentation were conducted in the study. A number of metrics were collected based on four variables:

- True positive (TP) – volume of correctly predicted lesion presence

- True negative (TN) – volume of correctly predicted lesion absence

- False positive (FP) – volume of incorrectly predicted lesion presence

- False negative (FN) - volume of incorrectly predicted lesion absence

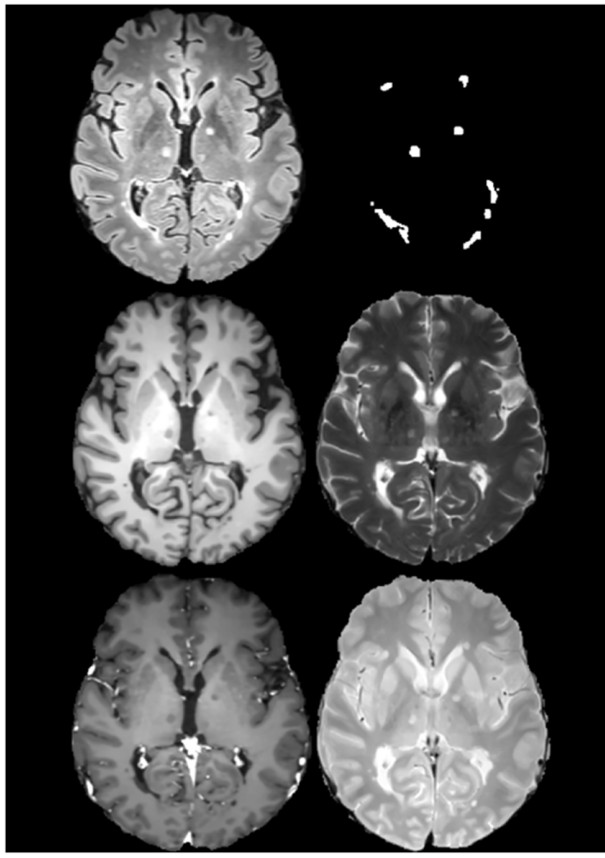

**Fig 2. A sample collection (Patient 1 training dataset) of one MRI scan captured in the horizontal plane from the MICCAI 2016 MS dataset of the same patient using different imaging protocols [27].** From left to right: top row, **(a)** FLAIR and **(b)** Consensus position of lesions based upon the manual delineation of seven radiologists; middle row, **(c)** T1 and **(d)** T2; bottom row, **(e)** gadolinium-enhanced and **(f)** proton density. Permission to use these images was provided by Centre de coordination de l'OFSEP, Hôpital neurologique Pierre Wertheimer, Service de neurologie A, in accordance with the data access agreement for the data-set used [36,39].

Firstly Dice, given by

$$Dice = \frac{2TP}{2TP + FP + FN} \qquad (1)$$

TPR (sensitivity),

$$TPR = \frac{TP}{TP + FN}, \qquad (2)$$

FPR (fall-out rate)

$$FPR = \frac{FP}{FP + TN}, \qquad (3)$$

FNR (miss rate)

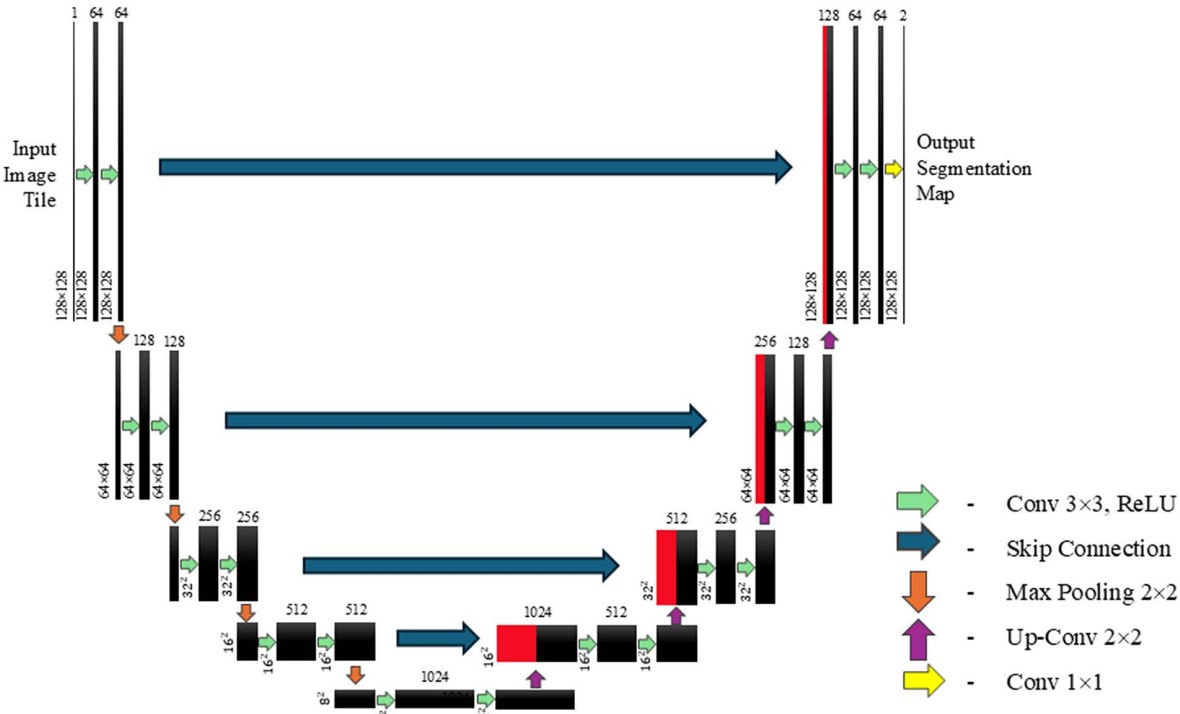

**Fig 3. U-Net architecture: blue boxes represent multi-channel feature map (original image).** The number of channels are indicated by the number above the box. White boxes represent replicated feature maps. Arrows represent specific operations utilised. Note that unlike in the original Ronneberger diagram, padding is used to keep the model dimensionally consistent in the upsampling and downsampling [16].

$$FNR = \frac{FN}{FN + TP},$$

(4)

and TNR (specificity)

$$TNR = \frac{TN}{TN + FP}.$$

(5)

Across all substudies described in the methodology, each test permutation was run at least five times. Latterly, the inclusion of additional metrics allowed for a more thorough analysis of model performance on the dataset. The primary metric was Dice since it directly assesses the similarity between ground truth and prediction. The sequence of data in the training set was randomly shuffled at the initialisation of each run to prevent the neural network from learning the ordering of the data, and hence promote its generalisation ability.

An initial sub-study was conducted involving the comparison of lesion detection performance of U-Net using different MRI sequences: namely: FLAIR, T1, T2, gadolinium-enhanced images and proton density images (Fig 2). The five tests in this sub-study acted as a controls to gauge the impact of changes and augmentations.

The effect of using multiple image modalities in combination on the ability of U-Net to detect MS lesions was explored. To limit the number of permutations (3,000 possible runs), five two-level, five-factor Taguchi Design of Experiments (DoE) were created, each using different combinations of image types for training, and one of the five image modalities to test the result of the training procedure [40]. This design led to eight permutations per DOE each of which was run five times,

limiting the number of proposed runs to 200 (8×5×5). The Taguchi DoE format is depicted in Table 2: a ✓ indicates the utilisation of an image sequence and a ×, its non-use. All tests were run five times and the mean was taken. For this section of the study, it was necessary to create control tests in which data was duplicated to ensure that when the neural network was simultaneous trained on the T1, T2 and FLAIR data (training dataset size of 45) for example, it could be compared to a version of the algorithm trained on the same number of images (e.g., 3 × 15 FLAIR images) without excluding any of the data.

Data augmentation was explored and since convolutional neural networks are not rotationally invariant, investigation of data expansion using Monai orientation transforms was carried out [41]. It is possible to expand the number of images for training by simply rotating them. The objective of doing this was to examine whether a larger dataset would improve accuracy with lesion segmentation, by how much and which types of transforms are most effective in achieving this. During this phase, duplicating images from the training set (for the control) was essential to ensure equal dataset sizes for evaluating the effects of employing multiple image modalities in a run. Examples of these rotations are illustrated in Fig 4, using a FLAIR image in the sagittal plane. The permutations investigated are depicted in Table 3.

An assessment of how segmentation accuracy is affected by exclusion of lesions based on size was carried out. By editing the masks provided, holes below, above or within a certain range can be eliminated or preserved in the preprocessing phase. To achieve this, the volume of every lesion identified in the aggregate masks was calculated using a script, so that they could be separated into quartiles based on size. A second script was written to fill in holes based on the condition discovered using the first script. Although it was stated in the study accompanying the dataset, that the smallest lesions considered were limited to 3 mm$^3$ [27], the script created for this purpose found lesions as small as 1.38 mm$^3$. The explanation is most likely related to the irregularity in the shape of lesions or in the programmatic libraries used. Fig 5 shows the ground truth mask in the sagittal plane for patient 11 (training dataset), 118.12 mm$^3$ from the central cut (seen in Slicer). The first image shows Q1 in isolation (top left), Q1 and Q2 together (top right), Q1, Q2 and Q3 (bottom left) and then the original mask, Q1, Q2, Q3 and Q4 (bottom right). The rationale for the quartile distribution was that 2047 continuous lesion islands were present across the 53 subjects in the dataset varying in size from 1.38 mm$^3$ to 66300.60 mm$^3$. This was determined using a python script used to count and quantify the lesions through the processing of the lesion masks within dataset. The lesions were first ranked in ascending order by size across all patients. The smallest 512 lesions were assigned to Q1, the next 512 lesions (i.e., the 513th to 1024th lesions) were assigned to Q2, and subsequent quartiles were defined analogously. Lesion quartile assignment was performed on the combined dataset, including both training and testing data, rather than treating them independently. The assessments performed which excluded specific

**Table 2. Taguchi Design of Experiment (DoE) for image modality testing. Each permutation involved utilising different combinations of the training images, these were then tested against each testing set. Each DoE was repeated for each of the five testing sets. A tick indicates the utilisation of an image sequence and a ×, its non-use.**

**Taguchi Design of Experiment**

| Permutation | FLAIR | T1 | T2 | Gadolinium-enhanced | PD |
|---|---|---|---|---|---|
| No. 1 | ✓ | ✓ | ✓ | ✓ | ✓ |
| No. 2 | ✓ | ✓ | ✓ | × | × |
| No. 3 | ✓ | × | × | ✓ | ✓ |
| No. 4 | ✓ | × | × | × | × |
| No. 5 | × | ✓ | × | ✓ | × |
| No. 6 | × | ✓ | × | × | ✓ |
| No. 7 | × | × | ✓ | ✓ | × |
| No. 8 | × | × | ✓ | × | ✓ |

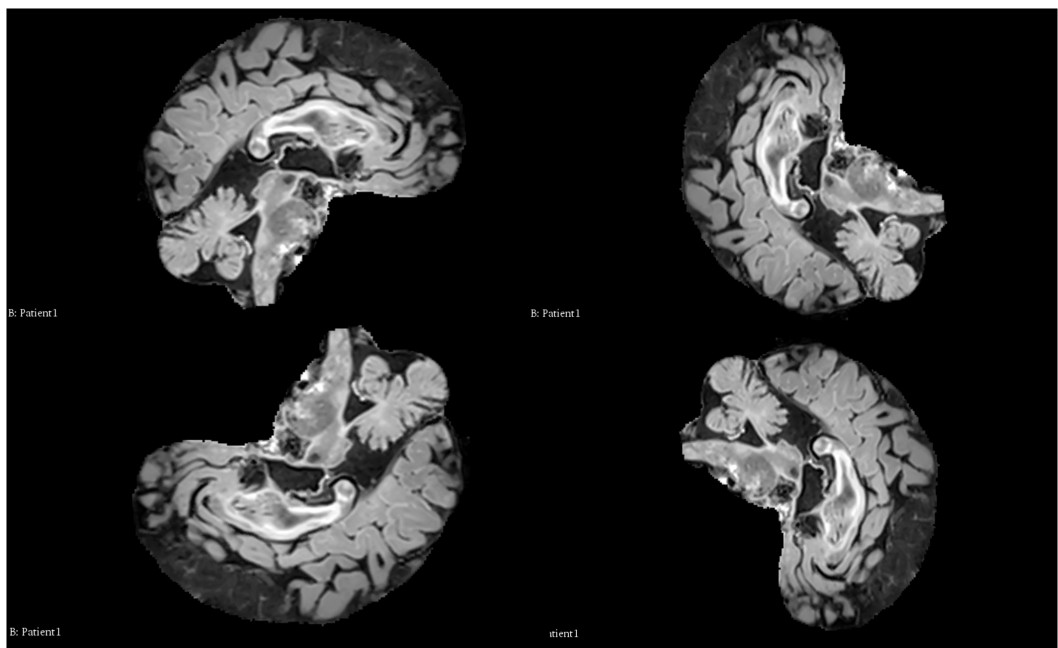

**Fig 4. A depiction of the rotations used for data augmentation visualised using 3D slicer software.** Patient 1 of the training dataset is shown in the sagittal plane using the FLAIR modality. Top row from left to right: the original image and 90° rotation. Bottom row from left to right: 180° rotation and 270° rotation. Permission to use these images was provided by Centre de coordination de l'OFSEP, Hôpital neurologique Pierre Wertheimer, Service de neurologie A, in accordance with the data access agreement for the data-set used [36,39].

**Table 3. Planned augmentation permutations. A tick indicates the use of a dataset and a cross its omission.**

**Rotation Transformations**

|  | original | 90 | 180 | 270 |
|---|---|---|---|---|
| No. 1 | ✓ | × | × | × |
| No. 2 | ✓ | ✓ | × | × |
| No. 3 | ✓ | ✓ | ✓ | × |
| No. 4 | ✓ | ✓ | ✓ | ✓ |

quartiles are detailed in Table 4. The lesion size analysis was limited to two MRI modalities, FLAIR and proton density images to limit the number of runs performed.

The best permutation of each of the substudies, based on best Dice output, was combined to maximise segmentation results. During all assessments the components of Dice in the context of lesion detection were evaluated; namely TPR, TNR, FPR and FNR. Patient 26 was also removed from the testing set as its ground truth was absent within the files. This provided an opportunity to examine the impact of removing defective data on segmentation.

The ability to train U-Net based on one imaging modality and then test it on other imaging modalities was explored. U-Net was independently trained on each imaging modality, and then tested with the remaining imaging modalities. All 20 permutations were individually evaluated (Table C in S2 Text). A further five control simulations were completed, with controls conducted for each permutation. Along with the controls which were repeated due to a significant upgrade to the Bluebear system on which the simulations had been run.

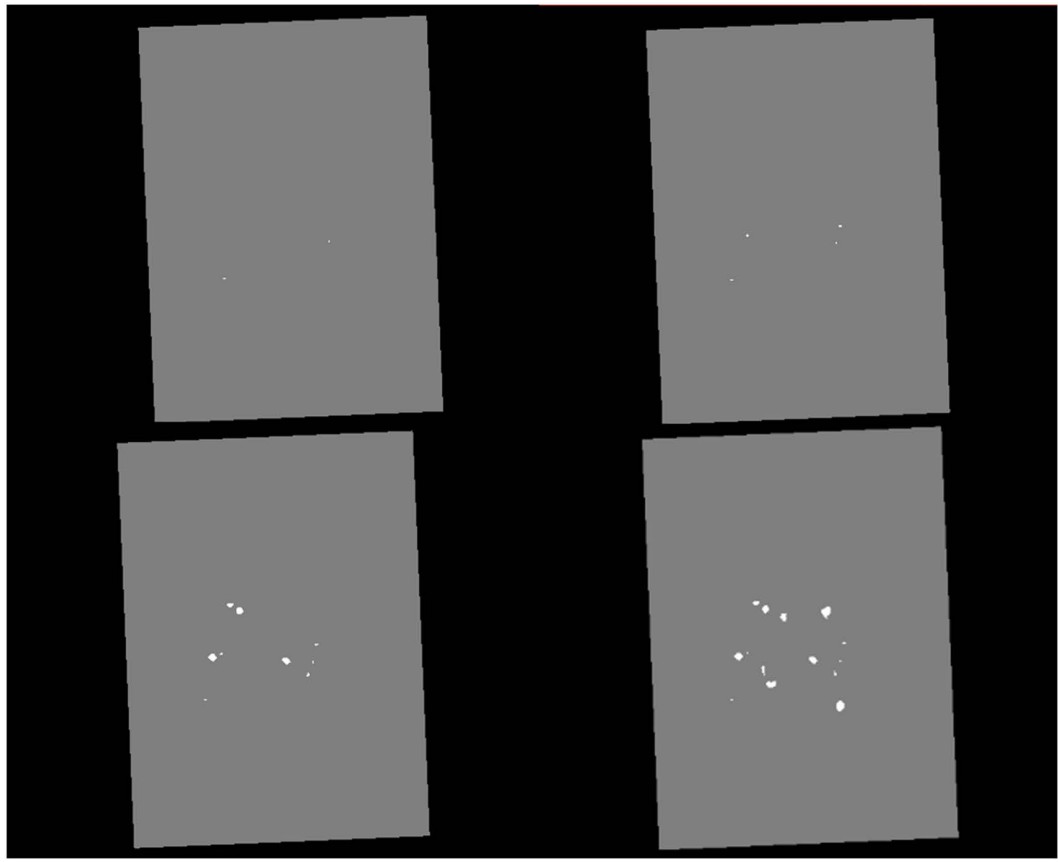

**Fig 5. An illustration of how limiting lesion size impacts the ground truth mask visually - rendered using 3D Slicer software.** The ground truth mask for patient 11 of the training dataset is shown in the axial plane. Top row from left to right: Q1 (1.38 - 13.56 mm³) and Q1-Q2 (1.38 - 36.43 mm³). Bottom row from left to right: Q1-Q3 (1.38 - 100.46 mm³) and Q1-Q4 (original mask). Permission to use these images was provided by Centre de coordination de l'OFSEP, Hôpital neurologique Pierre Wertheimer, Service de neurologie A, in accordance with the data access agreement for the data-set used [36,39].

**Table 4. Lesion quartiles analysis indicating which lesions were and were not used for each of the tests undertaken. The tick mark indicates the use of a training set, the cross indicates that it was not used.**

**Lesion Size Analysis**

|  | Q1 | Q2 | Q3 | Q4 |
|---|---|---|---|---|
| Lesion volume mm³ | 1.38 - 13.56 | 13.56 - 36.43 | 36.43 - 100.46 | 100.46 - 66300.60 |
| No. 1 | ✓ | × | × | × |
| No. 2 | ✓ | ✓ | × | × |
| No. 3 | ✓ | ✓ | ✓ | × |
| No. 4 | × | ✓ | × | × |
| No. 5 | × | ✓ | ✓ | × |
| No. 6 | × | ✓ | ✓ | ✓ |
| No. 7 | × | × | ✓ | × |
| No. 8 | × | × | ✓ | ✓ |
| No. 9 | × | × | × | ✓ |

Mann-Whitney U-tests were used to ascertain the statistical significance of the obtained differences in Dice scores. Kruskal-Wallis tests were also used to assess the statistical significance between control tests (Table 6), in which multiple versions of the same data were used. This allowed for the impact of a data modification to be determined. If the resulting p-value was less than 0.05, it indicates that the two sets of data were likely statistically different from one another.

All experimentation was carried out using High-Performance Computing facilities (Bluebear, located at the University of Birmingham), using a single GPU node and all tests were allowed 2000 epochs to run to ensure best results possible could be achieved for any one permutation.

## 3. Results

Initial control testing was performed on each of the five original MRI datasets. No data augmentations were included to find a point of reference for future experimentation. Equations 1, 2, 3, 4 and 5 are the metrics used to assess the U-Net outputs. Dice score results ranged from 0.576 (FLAIR) to 0.571 (Gado). Using Kruskal-Wallis analysis on the Dice results, the difference between the scores proved to be statistically significant, with a p-value of 0.0065, however when excluding the results associated with the gadolinium-enhanced dataset, the p-value of 0.7402 indicated no significant statistical difference in Dice, suggesting that the gadolinium-enhanced dataset was an outlier. The average epochs to achieve highest Dice scores were between 1544 and 1610 (Table 5), meaning that the 2000 epochs used were excessive for this phase of experimentation and resulted in overfitting to the training datasets.

Utilising Kruskal-Wallis, it was found that in the majority of cases there were no statistically significant differences between the controls doubling, tripling, quadrupling or quintupling ($p \leq 0.05$) the dataset using the same data over 2000 epochs (Table 6). The exception to this was the proton density images.

Mann-Whitney U-tests were used to compare results based on the Taguchi Design of Experiment and appropriate controls. Analysis indicated that the majority of outcomes were not statistically significant or performed worse than their

**Table 5. Mean results for initial testing of each of the MRI modalities over 10 runs.**

| Results For Phase 1 | | | | | |
|---|---|---|---|---|---|
| Modality | T1 | T2 | Gado | PD | FLAIR |
| Best mean Dice | 0.5747 | 0.5744 | 0.5717 | 0.5717 | 0.5753 |
| Min (Dice) | 0.5716 | 0.5712 | 0.5686 | 0.5727 | 0.5723 |
| q1 (Dice) | 0.5739 | 0.5728 | 0.5704 | 0.574 | 0.5739 |
| Median (Dice) | 0.5743 | 0.5742 | 0.5717 | 0.5757 | 0.5751 |
| q3 (Dice) | 0.5759 | 0.5759 | 0.5730 | 0.5764 | 0.5763 |
| Max (Dice) | 0.5774 | 0.5781 | 0.5744 | 0.5791 | 0.5819 |
| Runs | 10 | 10 | 10 | 10 | 10 |
| Maximum Dice | 0.5774 | 0.5781 | 0.5744 | 0.5791 | 0.5819 |
| Minimum Average Dice | 0.5716 | 0.5712 | 0.5686 | 0.5757 | 0.5723 |
| Mean Epoch to best mean Dice | 1601 | 1544 | 1571 | 1610 | 1619 |
| Best Average TPR | 0.9651 | 0.9648 | 0.9639 | 0.9646 | 0.9304 |
| Epoch at best Average TPR | 98 | 101 | 94 | 95 | 76 |
| Best Average FPR | 0.0574 | 0.0573 | 0.0577 | 0.0575 | 0.0583 |
| Epoch at best Average FPR | 1997 | 1993 | 1999 | 1996 | 1999 |
| Best Average FNR | 0.0349 | 0.0352 | 0.0361 | 0.0354 | 0.0696 |
| Epoch at best Average FNR | 98 | 101 | 94 | 95 | 76 |
| Best Average TNR | 0.9426 | 0.9427 | 0.9423 | 0.9425 | 0.9417 |
| Epoch at best Average TNR | 1997 | 1999 | 1999 | 1996 | 1999 |

**Table 6. Control testing for each of the MRI modalities. The control number represents the factor by which the dataset has been multi-plied so that direct comparison can be made with enlarged augmented datasets. Statistically significance of results was assessed using Kruskal-Wallis (K-W) tests.**

**Controls Testing**

| Modality | T1 | T2 | Gado | PD | FLAIR |
|---|---|---|---|---|---|
| Control 1 | 0.5747 | 0.5744 | 0.5717 | 0.5717 | 0.5753 |
| Control 2 | 0.5826 | 0.5860 | 0.5848 | 0.5845 | 0.5730 |
| Control 3 | 0.5813 | 0.5802 | 0.5819 | 0.5799 | 0.5689 |
| Control 4 | 0.5818 | 0.5836 | 0.5771 | 0.5831 | 0.5722 |
| Control 5 | 0.5779 | 0.5818 | 0.5771 | 0.5768 | 0.5717 |
| Range | 0.0079 | 0.0116 | 0.0131 | 0.0128 | 0.0064 |
| K-W Score | 0.4164 | 0.1076 | 0.0539 | 0.0163 | 0.1395 |

respective controls (Table 7), implying that dataset hybridisation of multiple MRI sequences for this use was not benefi-cial in improving lesion segmentation. One of the more interesting results was that training using the FLAIR dataset, and testing using all other imaging modalities was more effective than the training and testing on the same data type (Taguchi DoEs 2 3 4 5 of Experiment 4). Using a single rotation produced a statistically significant improvement for using the FLAIR dataset against the associated control (p-value = 0.0317) as analysed by a Mann-Whitney U-test. It was found, however, that no further benefits were gained from duplicating and then rotating the dataset and, in fact, performance diminished as additional rotation based augmentation was introduced to the training process. The decline in results was more pro-nounced using the proton density images, especially when further augmentations were included. The mean Dice for pro-ton density images dropped off from 0.5862 to 0.5785 illustrating the limitations of augmentation of this kind in the context of MS lesion segmentation (Table 8).

The rubric set in this challenge was ignored in the next phase of experimentation, in which the impact of lesion size on the detection process was examined. Segmentation results for smaller lesions in isolation were significantly worse than those of the controls (Table 8). Exclusion of lesions < 36 mm³ however, did lead to a statistically significant increase in Dice for both proton density and FLAIR testing. In all cases, the data with smaller lesions excluded in the ground truth, pro-duced better results than the respective controls.

Inverting testing and training produced improved results relative to controls in all cases achieving a level of statis-tical significance. The p-values produced against each of the respective controls were statistically significant ($0.0027 \leq p \leq 0.0067$) for every image type. In every instance, the mean Dice increased (Table 9), by up to 2.8% (in the case of the gadolinium images). Redistributing the training and testing data (43:10) to increase the size of the latter failed to improve the segmentation accuracy (Table A in S2 Text). In three of the five cases, results were better than the respec-tive control tests but none exceeded any of the previous set of tests, in which the training to testing distribution was 38:15.

When the volume containing the missing ground truth was omitted, Dice scores improved across all modalities, the improvements of which were statistically significant in all cases ($0.0002 \leq p \leq 0.0036$) against their respective controls (Table B in S2 Text). Although all Dice scores increase, FLAIR results went from being the best to the worst-performing of all image modalities tested, improving only by 0.60%, suggesting that FLAIR is the modality least sensitive to errors. The TPR and TNR with best Dice scores were 0.221 and 0.927 respectively. They also proved to be the worst-performing image modalities. Proton density images were the best performing modality with a Dice of 0.590. The TPR and TNR asso-ciated with the proton density dataset were 0.241 and 0.935 respectively, in which the latter result was the best TNR and the TPR was only surpassed by the T2 TNR result of 0.247. Proton density, T1, T2 and gadolinium-enhancement lesion detection rates increased by an average of 1.63%, 1.38%, 1.45% and 1.42% respectively.

**Table 7. Mean Dice results of the Taguchi experiments against relevant controls. Each test was run five times with each of the results displayed (min, q1, med, q3 and max).**

**Results For Phase 2**

| | Dice | Control | p-value | p ≤ 0.05 | Min | q1 | Med | q3 | Max |
|---|---|---|---|---|---|---|---|---|---|
| Taguchi 1 (using FLAIR testing) | | | | | | | | | |
| Experiment 1 | 0.5689 | 0.5717 | 0.0318 | ✓ | 0.5679 | 0.5682 | 0.5683 | 0.5689 | 0.5710 |
| Experiment 2 | 0.5708 | 0.5689 | 0.3095 | × | 0.5666 | 0.5671 | 0.5719 | 0.5738 | 0.5745 |
| Experiment 3 | 0.5723 | 0.5689 | 0.1508 | × | 0.5680 | 0.5684 | 0.5729 | 0.5760 | 0.5762 |
| Experiment 4 | 0.5786 | 0.5757 | 0.0500 | × | 0.5772 | 0.5773 | 0.5787 | 0.5798 | 0.5802 |
| Experiment 5 | 0.5620 | 0.5731 | 0.0080 | ✓ | 0.5590 | 0.5603 | 0.5616 | 0.5633 | 0.5656 |
| Experiment 6 | 0.5690 | 0.5731 | 0.3095 | × | 0.5614 | 0.5692 | 0.5696 | 0.5697 | 0.5753 |
| Experiment 7 | 0.5709 | 0.5731 | 0.6905 | × | 0.5675 | 0.5689 | 0.5719 | 0.5728 | 0.5735 |
| Experiment 8 | 0.5601 | 0.5731 | 0.0079 | ✓ | 0.5560 | 0.5584 | 0.5596 | 0.5600 | 0.5664 |
| Taguchi 2 (using T1 testing) | | | | | | | | | |
| Experiment 1 | 0.5780 | 0.5779 | 1.000 | × | 0.5750 | 0.5777 | 0.5780 | 0.5794 | 0.5799 |
| Experiment 2 | 0.5844 | 0.5813 | 0.4206 | × | 0.5806 | 0.5827 | 0.5831 | 0.5868 | 0.5887 |
| Experiment 3 | 0.5819 | 0.5813 | 0.8413 | × | 0.5779 | 0.5790 | 0.5811 | 0.5847 | 0.5867 |
| Experiment 4 | 0.5896 | 0.5747 | 0.0007 | ✓ | 0.5882 | 0.5888 | 0.5899 | 0.5904 | 0.5908 |
| Experiment 5 | 0.5843 | 0.5826 | 0.4206 | × | 0.5820 | 0.5831 | 0.5850 | 0.5852 | 0.5863 |
| Experiment 6 | 0.5871 | 0.5826 | 0.1161 | × | 0.5833 | 0.5834 | 0.5877 | 0.5900 | 0.5913 |
| Experiment 7 | 0.5882 | 0.5826 | 0.1508 | × | 0.5810 | 0.5870 | 0.5895 | 0.5909 | 0.5927 |
| Experiment 8 | 0.5857 | 0.5826 | 0.6905 | × | 0.5772 | 0.5808 | 0.5855 | 0.5915 | 0.5933 |
| Taguchi 3 (using T2 testing) | | | | | | | | | |
| Experiment 1 | 0.5783 | 0.5818 | 0.0556 | × | 0.5751 | 0.5774 | 0.5780 | 0.5794 | 0.5816 |
| Experiment 2 | 0.5817 | 0.5802 | 0.5476 | × | 0.5793 | 0.5797 | 0.5817 | 0.5835 | 0.5841 |
| Experiment 3 | 0.5861 | 0.5802 | 0.0952 | × | 0.5800 | 0.5825 | 0.5872 | 0.5900 | 0.5909 |
| Experiment 4 | 0.5898 | 0.5744 | 0.0027 | ✓ | 0.5870 | 0.5882 | 0.5899 | 0.5910 | 0.5927 |
| Experiment 5 | 0.5838 | 0.5860 | 0.3457 | × | 0.5806 | 0.5814 | 0.5847 | 0.5858 | 0.5867 |
| Experiment 6 | 0.5900 | 0.5860 | 0.4206 | × | 0.5829 | 0.5832 | 0.5898 | 0.5964 | 0.5976 |
| Experiment 7 | 0.5873 | 0.5860 | 0.6905 | × | 0.5815 | 0.5845 | 0.5858 | 0.5913 | 0.5942 |
| Experiment 8 | 0.5823 | 0.5860 | 0.1508 | × | 0.5785 | 0.5804 | 0.5826 | 0.5846 | 0.5855 |
| Taguchi 4 (using PD testing) | | | | | | | | | |
| Experiment 1 | 0.5813 | 0.5768 | 0.0556 | × | 0.5764 | 0.5815 | 0.5819 | 0.5829 | 0.5840 |
| Experiment 2 | 0.5838 | 0.5799 | 0.1508 | × | 0.5821 | 0.5823 | 0.5825 | 0.5828 | 0.5891 |
| Experiment 3 | 0.5822 | 0.5799 | 0.3457 | × | 0.5764 | 0.5819 | 0.5829 | 0.5845 | 0.5854 |
| Experiment 4 | 0.5889 | 0.5753 | 0.0007 | ✓ | 0.5881 | 0.5882 | 0.5889 | 0.5890 | 0.5901 |
| Experiment 5 | 0.5833 | 0.5852 | 0.8412 | × | 0.5777 | 0.5806 | 0.5851 | 0.5859 | 0.5872 |
| Experiment 6 | 0.5833 | 0.5852 | 0.834 | × | 0.5797 | 0.5812 | 0.5849 | 0.5854 | 0.5855 |
| Experiment 7 | 0.5858 | 0.5852 | 0.4633 | × | 0.5831 | 0.5840 | 0.5848 | 0.5874 | 0.5899 |
| Experiment 8 | 0.5824 | 0.5852 | 0.3095 | × | 0.5796 | 0.5804 | 0.5833 | 0.5842 | 0.5845 |
| Taguchi 5 (using Gado testing) | | | | | | | | | |
| Experiment 1 | 0.5843 | 0.5811 | 0.4206 | × | 0.5788 | 0.5814 | 0.5822 | 0.5855 | 0.5935 |
| Experiment 2 | 0.5835 | 0.5819 | 0.5296 | × | 0.5798 | 0.5811 | 0.5832 | 0.5843 | 0.5891 |
| Experiment 3 | 0.5838 | 0.5819 | 0.5296 | × | 0.5785 | 0.5830 | 0.5845 | 0.5858 | 0.5874 |
| Experiment 4 | 0.5892 | 0.5717 | 0.0007 | ✓ | 0.5881 | 0.5885 | 0.5891 | 0.5896 | 0.5907 |
| Experiment 5 | 0.5853 | 0.5848 | 1.000 | × | 0.5816 | 0.5831 | 0.5832 | 0.5857 | 0.5931 |
| Experiment 6 | 0.5881 | 0.5848 | 0.4206 | × | 0.5853 | 0.5865 | 0.5866 | 0.5890 | 0.5932 |
| Experiment 7 | 0.5847 | 0.5848 | 0.9166 | × | 0.5828 | 0.5832 | 0.5833 | 0.5839 | 0.5901 |

*(Continued)*

**Table 7.** (Continued)

**Results For Phase 2**

| | Dice | Control | p-value | p ≤ 0.05 | Min | q1 | Med | q3 | Max |
|---|---|---|---|---|---|---|---|---|---|
| Experiment 8 | 0.5842 | 0.5848 | 1.000 | × | 0.5780 | 0.5791 | 0.5851 | 0.5880 | 0.5908 |

**Table 8.** Segmentation results when rotational augmentations are included and results showing the impact of limiting lesion size. Lesion sizes are separated into quartiles. Q1 - (1.38-13.56 mm³). Q2 - (13.56-36.43 mm³). Q3 - (36.43-100.46 mm³). Q4 - (100.46-66300.60 mm³). As with the previous set of results, each test was run five times.

**Results For Phases 3 And 4**

| Experiment | Dice | Control | p-value | p≤0.05 | Min | q1 | Med | q3 | Max |
|---|---|---|---|---|---|---|---|---|---|
| FLAIR Testing for Phase 3 - Rotation Augmentations | | | | | | | | | |
| Orig+90° | 0.5809 | 0.5731 | 0.0317 | ✓ | 0.5760 | 0.5761 | 0.5810 | 0.5856 | 0.5858 |
| Orig+90°+180° | 0.5723 | 0.5689 | 0.1507 | × | 0.5677 | 0.5699 | 0.5710 | 0.5758 | 0.5773 |
| Orig+90°+180° +270° | 0.5728 | 0.5722 | 0.9166 | × | 0.5640 | 0.5709 | 0.5737 | 0.5776 | 0.5779 |
| PD Testing for Phase 3 - Rotation Augmentations | | | | | | | | | |
| Orig+90° | 0.5862 | 0.5845 | 0.2903 | × | 0.5766 | 0.5773 | 0.5782 | 0.5816 | 0.5873 |
| Orig+90°+180° | 0.5787 | 0.5799 | 0.0159 | ✓ | 0.5742 | 0.5776 | 0.5780 | 0.5814 | 0.5825 |
| Orig+90°+180° +270° | 0.5785 | 0.5831 | 0.1508 | × | 0.5731 | 0.5764 | 0.5773 | 0.5811 | 0.5847 |
| FLAIR Testing for Phase 4 - Lesion Size Analysis | | | | | | | | | |
| Q1 | 0.5050 | 0.5757 | 0.0007 | ✓ | 0.5045 | 0.5047 | 0.5049 | 0.5050 | 0.5056 |
| Q1+Q2 | 0.5060 | 0.5757 | 0.0007 | ✓ | 0.5057 | 0.5059 | 0.5060 | 0.5061 | 0.5062 |
| Q1+Q2+Q3 | 0.5093 | 0.5757 | 0.0007 | ✓ | 0.5090 | 0.5091 | 0.5092 | 0.5096 | 0.5097 |
| Q2 | 0.5056 | 0.5757 | 0.0007 | ✓ | 0.5051 | 0.5051 | 0.5058 | 0.5061 | 0.5061 |
| Q2+Q3 | 0.5086 | 0.5757 | 0.0007 | ✓ | 0.5081 | 0.5084 | 0.5086 | 0.5087 | 0.5094 |
| Q2+Q3+Q4 | 0.5790 | 0.5757 | 0.0431 | ✓ | 0.5775 | 0.5782 | 0.5782 | 0.5791 | 0.5820 |
| Q3 | 0.5082 | 0.5757 | 0.0007 | ✓ | 0.5081 | 0.5082 | 0.5082 | 0.5082 | 0.5084 |
| Q3+Q4 | 0.5787 | 0.5757 | 0.1645 | × | 0.5741 | 0.5773 | 0.5791 | 0.5794 | 0.5837 |
| Q4 | 0.5807 | 0.5757 | 0.0199 | ✓ | 0.5796 | 0.5798 | 0.5807 | 0.5812 | 0.5823 |
| PD Testing for Phase 4 - Lesion Size Analysis | | | | | | | | | |
| Q1 | 0.5050 | 0.5726 | 0.0027 | ✓ | 0.5048 | 0.5049 | 0.5050 | 0.5052 | 0.5052 |
| Q1+Q2 | 0.5073 | 0.5726 | 0.0026 | ✓ | 0.5069 | 0.5073 | 0.5073 | 0.5076 | 0.5076 |
| Q1+Q2+Q3 | 0.5100 | 0.5726 | 0.0027 | ✓ | 0.5096 | 0.5099 | 0.5102 | 0.5102 | 0.5103 |
| Q2 | 0.5059 | 0.5726 | 0.0007 | ✓ | 0.5056 | 0.5057 | 0.5059 | 0.5060 | 0.5061 |
| Q2+Q3 | 0.5087 | 0.5726 | 0.0027 | ✓ | 0.5085 | 0.5085 | 0.5086 | 0.5089 | 0.5091 |
| Q2+Q3+Q4 | 0.5800 | 0.5726 | 0.0027 | ✓ | 0.5775 | 0.5800 | 0.5801 | 0.5803 | 0.5822 |
| Q3 | 0.5077 | 0.5726 | 0.0267 | ✓ | 0.5071 | 0.5071 | 0.5073 | 0.5076 | 0.5093 |
| Q3+Q4 | 0.5855 | 0.5726 | 0.0007 | ✓ | 0.5826 | 0.5837 | 0.5853 | 0.5876 | 0.5884 |
| Q4 | 0.5830 | 0.5726 | 0.0007 | ✓ | 0.5808 | 0.5824 | 0.5837 | 0.5838 | 0.5842 |
| Final Testing - PD Used | | | | | | | | | |
| Q3+Q4 Orig+90° | 0.5896 | 0.5726 | 0.0007 | ✓ | 0.5864 | 0.5885 | 0.5886 | 0.5908 | 0.5919 |

**Table 9. Results using 2016 MICCAI MS Lesion Segmentation Dataset. DSC = Dice; TPR = True Positive Rate; PPV = Positive Predictive Value.**

| Method | DSC | TPR | PPV | Data type(s) used |
|---|---|---|---|---|
| [42] | 0.566 | 0.456 | 0.287 | FLAIR, T1, T2 |
| [13] | 0.571 | 0.358 | 0.526 | FLAIR, T2 |
| [43] | 0.60 | 0.53 | 0.80 | FLAIR |
| [14] | 0.502 | × | × | FLAIR |
| [44] | 0.651 | 0.689 | × | T1, FLAIR |
| [45][a]* | 0.63 | 0.57 | × | FLAIR |
| [45][b]* | 0.60 | 0.69 | × | FLAIR |
| this study | 0.59 | 0.27 | × | PD |
| [24]* | 0.866 | 0.856 | × | T1, T2, FLAIR |
| [26]* | 0.824 | 0.762 | × | Not explicitly stated |
| [25]* | 0.76 | 0.65 | × | Concatenation of all 5 modalities |

[a]LeMan-PV method.

[b]PV-CNN method.

*Studies that use dataset but do not adhere to the set rubric of the challenge.

All of the most effective data adaptions were applied to the dataset which resulted in the best Dice scores and then tested. This involved using the proton density images, incorporating a single rotation dataset to double the training set size and restricting lesion sizes to the two larger quartiles (Table 8). The Dice score achieved was 0.590 across an average of 10 runs. All results of interest were plotted in Fig 6, which indicates the average best Dice score for a particular permutation against the average epoch at which it was achieved. Between five and ten runs were carried out for each permutation depicted in the graph. An example of the segmentation is depicted in Fig 7. Note that a FLAIR image has been used as they are visually more informative to the human eye. The training and testing processes for the best results are illustrated in Fig 8. It was decided that 700 epochs would be used as it had been found that the best Dice score for this particular setup had occurred on average at epoch 619.

An additional set of results were collected to conclusively determine the impact of using different imaging modalities. Each of the imaging modalities were used for testing and training in isolation unlike in phase 2 in which datasets were used in combination. The majority of combinations showed little statistically significant difference against respective controls (Table C in S2 Text). The only exception was using the training FLAIR dataset in which Dice score declined by at least 1.6% in every instance relative to the control Dice score. In every other test, the differences between results and controls were no greater than 0.5%.

## 4. Discussion

This is the first investigation that comprehensively assesses the impact of data augmentation and manipulation in the context of detection of MS lesions by MRI. The main findings of this study were (1) that initial testing, using the entire dataset, indicated that FLAIR was the optimal image modality and gadolinium-enhanced images produced the lowest accuracy during the automated segmentation process of lesions (Table 5) with Dice scores of 0.5753 and 0.5717, respectively; (2), that the model could be trained on one group of MRI types and prove effective in automatically segmenting lesions in another (Table C in S2 Text); (3), inclusion of one single transformation in tandem for training resulted in an improvement in performance relative to the controls in all instances (Table 8); and (4) exclusion of the smallest lesions (lesions < 36 mm$^3$ in volume), produced higher Dice scores which were statistically significant compared to the relevant control. The set of results recorded using all the data changes and which produced the best outcomes described in the results section gave a Dice score of 0.589 (Table 8), comparable to studies which adhered to the rubric of the MICCAI 2016 challenge [13,14,42–45].

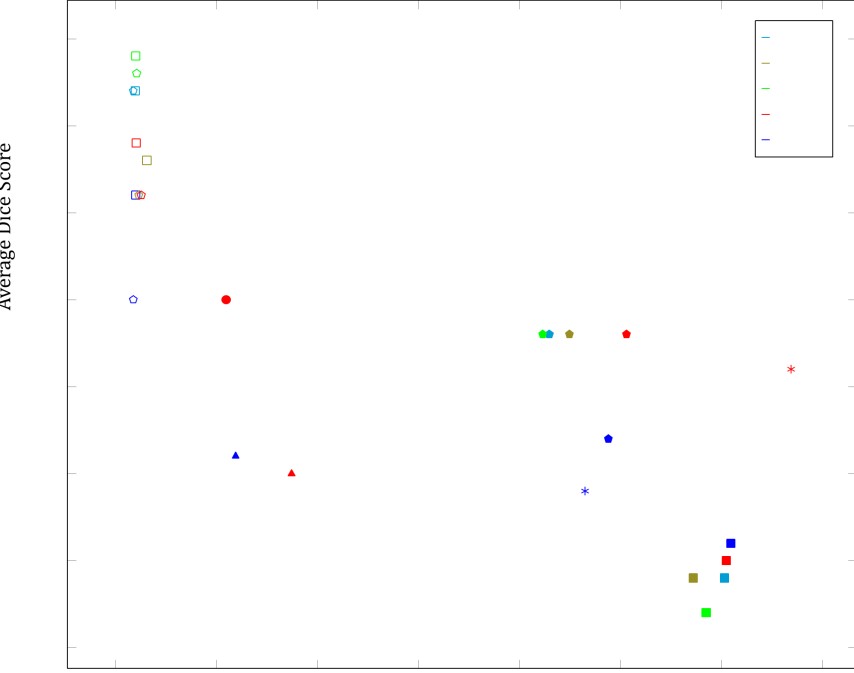

**Fig 6. A scatter graph illustrating average Dice scores achieved with the 2016 MICCAI dataset in this study using various augmentations and MRI modalities, plotted against the average epoch at which the score was achieved.** Initial control testing is indicated by ■. Inverted dataset testing is indicated by □. Best rotation results are indicated by ▲. Best results when lesion size limitation is implemented illustrated by *. Best result without altering dataset training and testing data distributions ●. Removal of image without ground truth from the data ⬟. Removal of image without ground truth from the data inverted ⬠.

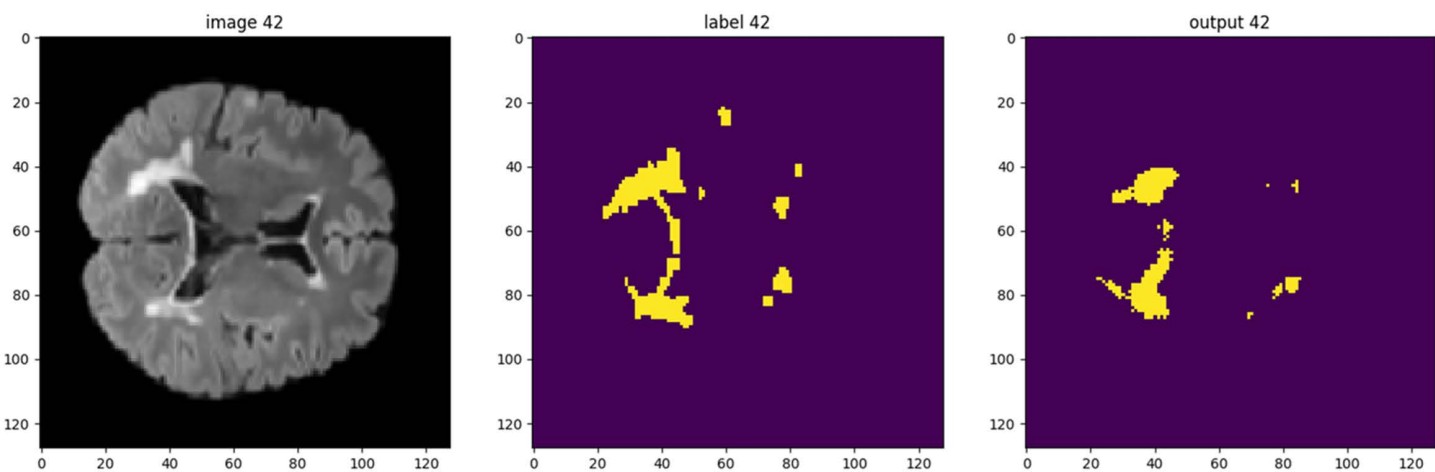

**Fig 7. An illustration of an MRI (left), its ground truth label indicating lesion location (centre) and the resulting trained U-Net prediction (right).** Permission to use these images (left and centre) was provided by Centre de coordination de l'OFSEP, Hôpital neurologique Pierre Wertheimer, Service de neurologie A, in accordance with the data access agreement for the data-set used [36,39].

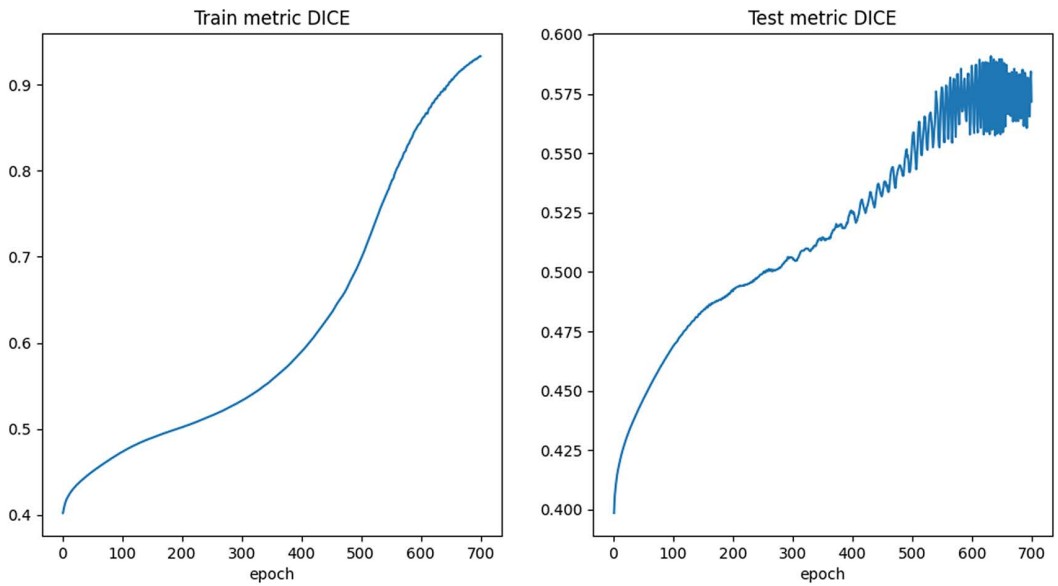

**Fig 8. A set of graphs illustrating the progressions of both the training and testing processes based on the Dice Metric from best results.**

## 4.1. Impact of model

Since CNNs are not rotationally invariant, they perceive the data from the same patient but rotated through 90°, 180° or 270° as distinct from its original with FLAIR. In this current study, it has been demonstrated that introducing additional augmentations improved segmentation illustrated by the Orig+90 FLAIR results in Table 8 where results Dice were found to improve from 0.5731 to 0.5809 when using the real dataset in combination with a single set of rotated images. However further augmentations not only failed to improve results but ended up diminishing them with each successive version. This may be a consequence of overfitting to data produced in the three data centres used for training (note that data produced in one of the four centres was exclusively used for testing). Another explanation for the limited impact of introducing rotations is that it only served to increase the complexity of the problem. While lesions can exhibit high variability in shape and spatial distribution, making them relatively invariant to canonical rotations, the anatomical structure of the brain is fixed and is not characterised by the same rotational invariance. Whilst rotations may benefit the model in segmenting lesions, they can also disrupt spatial consistency by introducing structurally dissimilar anatomy, which has the potential to add complexity to the learning process, causing the trained model less effective.

Another study used at least a portion of the dataset utilised and by using similar augmentations increased the Dice score by 6% [24]. In contrast, the augmentation undertaken in this study only increased segmentation quality by less than a percent, from 0.5731 to 0.5809 (Table 8). The likely reason for greater improvement is that the distribution between training and testing data was adjusted to a ratio of 4:1 and that one of the centres was excluded from the study. Furthermore, since the data was shuffled prior to the partition of training data and testing data, slices from the same volumes would almost certainly have been used for both training and testing which would unfairly benefit a model because near identical images which under normal circumstances would be linked would in this case be separated. This means that in many instances, nearly identical images would be used in the training and testing processes, something which would never happen in real-life medical environment. In the case of this work, images from the same volume were used either in training or testing putting the model used in this study at a disadvantage in comparison [24].

The information and analysis presented in this study, particularly regarding data preprocessing, the influence of MRI modalities, and data manipulation, are intended to serve as a benchmark for future research, providing a meaningful point of comparison for models used by other researchers. Further exploration could involve alternative neural network architectures, including variations of U-Net such as nnU-Net [18], as well as other segmentation models. Additional data augmentation strategies, such as test-time augmentation (TTA) [46], and the use of alternative datasets, including ISBI 2015 [34] and MICCAI 2021 [29], also merit investigation. Systematic comparison of benchmark results across different architectures, augmentation techniques, and datasets may help to isolate the contribution of each component to segmentation accuracy, thereby informing potential improvements to computational frameworks under development. While direct generalisation cannot be assumed, the findings of this study are nonetheless expected to be broadly applicable to more recent iterations of U-Net and other convolutional neural networks with similar architectural principles such as the encoder–decoder structure, skip connections and convolutional feature hierarchies.

### 4.2. Considerations of AI in a medical imaging context

Consistency in ground truth is vital for neural network training and testing. The accuracy of the ground truth is hard to evaluate, but other studies which utilised a dataset in which the creation of the ground truth was assisted by software alongside two experienced radiologists [28] as opposed to seven with purportedly less experience [31] presumably impacts performance. Errors in initial annotation can impact the whole study, both with the training and testing. Firstly, inaccuracies will affect the training process, as weights in the model adjust based on false information, generating false predictions in testing. Secondly, if inaccuracies are present in the ground truth of the test dataset, even if the training set is annotated accurately, assessment metrics like Dice will be low because the inaccurate ground truth of the testing data does not mirror predictions. The impact of errors in annotation was tangibly demonstrated by an inadvertently conducted section of this study, in which it was realised retrospectively that no ground truth had accompanied patient 26 of the testing data and that the ground truth volume was blank (Table B in S2 Text). The results improved substantially both when this volume was removed from the training and testing datasets. This can be seen firstly when comparing the sets of control data in Tables A and B in S2 Text for every imaging modality and their respective inversion results. Control testing and inverted testing were re-run yielding an improvement of results which was statistically significant across all imaging modalities. For example, segmentation of T1 volumes, simulated in line with competition rubric [31] improved from 0.5747 (Table A in S2 Text) to 0.5885 (Table B in S2 Text), similarly post inversion results for T1 images improved from 0.5970 (Table A in S2 Text) to 0.6013 (Table B in S2 Text). All results in these supplementary tables were an average of five runs and the improvement described in lesion segmentation for T1 volumes was consistent across all imaging modalities. Thirdly, the rubric outlined in the challenge associated with this dataset is unusual, demanding that only fifteen MRI volumes be used in training. Given that results improve after data is redistributed, it is clear that an insufficient amount of data is assigned to the training set for accurate segmentation results (Tables A and B in S2 Text).

There was found to be no increase in Dice using multiple MRI types in the training set (Table 7). It was hypothesised that additional scan types would act as extra information from which the neural network could learn. However, results from this current study suggest otherwise. An explanation for the lack of improvement may be a consequence of the neural network ultimately perceiving the differences between the image modalities as noise rather than distinctly different images. The fact that the controls either outperform the mixed dataset training or show no statistically significant improvement suggests that training on multiple datasets simultaneously offers no tangible benefit. Experiment 4 in all but FLAIR testing was the only notable exception but this was a unique case in which only one training modality was used (Table 7). Although not every image combination was tested, of the eight variations tried, none tested improved results when compared to the relevant control. A result of note from varying dataset modalities in training and testing in Table 7 was that in most cases there was no statistically significant difference between controls and circumstances in which the model was trained on FLAIR, gadolinium-enhanced and proton density images but tested on T1 and T2 images. This was subsequently

investigated in more detail, there was almost never any statistically significant difference between using a different modality in training than in testing. The only exceptions to this were when FLAIR imaging was used for testing. The implication of this is that there is potential for a universal model to be created, trained on FLAIR imaging, that is effective in identifying and segmenting lesions regardless of whether the image is a T1, T2, gadolinium-enhanced or proton density image (Table C in S2 Text). The similar lesion segmentation performance observed across MRI modalities has the potential to enable significant reductions in resource usage and financial cost, including MRI acquisition time and clinician workload, when monitoring lesion progression in individual patients. This may also improve diagnostic efficiency by reducing the need for multiple scans, thereby highlighting the potential utility of deploying machine learning–based segmentation models in hospital settings. Furthermore, machine learning models can be utilised to highlight patients with the most malignant abnormalities allowing for a medical facility to prioritise those most in need of urgent care more effectively. This is possible because the algorithms used are trained on datasets informed by the clinical expertise of medical professionals with many years of experience. This accumulated experience can then essentially be used to advise on treatment guidance and patient care, which has the potential to be especially beneficial to countries or areas with relatively few resources and experienced medical personnel.

Dataset size, variability and source also have implications on neural network performance. Smaller dataset size, higher variability and larger quantity data sources, such as scanners negatively impact the performance. The impact of these factors became evident when deviating from the rubric (Tables A and B in S2 Text). For training, 15 MS MRI volumes were gathered from three scanning centres, while 38 volumes from four centres were utilised for testing (including the three scanning centres previously used). When the datasets were swapped between training and testing tasks, segmentation results for all modalities demonstrated improvement. Furthermore, the algorithm had been trained using data from all the four test centres, while the testing data was from only three centres [31], and this was not the case for the control in which data from three centres was used for training and data from four centres was used for testing. A further set of tests were carried out to be sure that the algorithm could be both trained and tested on data produced in all centres. Interestingly, this produced worse results despite 43 MRI volumes being used in the training and only 10 in testing for all imaging modalities (e.g., FLAIR Dice reducing from 0.5891 to 0.5784 over an average of five runs each). This may reflect that the inclusion of data taken from additional centres in the training process impacts the testing process negatively. Whilst reducing the limiting dataset variability has positive impact on segmentation, an ideal scenario would be to create a model that can be universally applied to any brain scan and detect and segment objects of interest with high accuracy. The alternative to this would be to produce multiple models for individual scanning types which would also have drawbacks like needing to collect a large training dataset for each model – which is a major limiting factor in machine learning in general. This is especially problematic for medical studies in which there is often a limited supply of data because of the requirement for ethical approval for release and which generally take months to process. With the advent of generative techniques, expanding data size to produce novel synthetic data will allow much of the bureaucratic paperwork to be bypassed. In addition, generative machine learning models, like generative adversarial networks [47] and diffusion models [48] have the capacity to standardise imaging sets produced across different institutions, scanners and annotators. A method through which this can be achieved is image-to-image translation whereby a model learns to map between image domains [49]. This technique can be used to harmonise of the same species by, for instance, normalising resolution, adjusting contrast, or even converting one imaging modality to another. This results in improved consistency and interoperability across independently collected datasets. Synthetic data generation is an avenue which will be explored in future studies, in an effort to improve segmentation quality of MS lesions [50]. Nevertheless, this area of development remains in its infancy and benefits segmentation models to the same extent as image translations [51].

As with any machine learning study, there is always scope to include greater variability and increase the study numbers. However, to ensure a robust study a large dataset comprising 53 patients was chosen. Each of the volumes was made up of 512 2D slices meaning 27,136 (512 × 53) unique 2D images. Given that there are five versions of the dataset

due to the five imaging modalities, this amounts to 135,680 2D images. In terms of variability, the dataset was collected across four different centres using four differ MRI scanners using different protocols for each of modalities. In addition, the ground truth segmentation for the MICCAI 2016 MS dataset product of seven independent raters with relevant clinical training with exclusions made for outliers during the curation process when combining them. All of the aforementioned points make this particular dataset ideal for this study.

### 4.3. MS lesion segmentation specific issues when using AI

There are specific issues that affect MS lesion segmentation which are not seen in other medical imaging studies. One of the fundamental limitations in all of the sub-studies performed lies in the use of the Dice metric combined with the nature of the data used. The class imbalance within the data, namely lesion volume relative to the rest of the image being very small, and the Dice being calculated for both classes (lesions and absence of lesions) and then averaged, means that the loss function used, which is the derivative of the Dice, will prioritise the negative space. This is because the function aims to maximise the Dice score and the optimal way of doing this is to show bias towards the majority, as this has the greatest impact on minimising loss. This is fundamentally problematic in MS lesion segmentation, as the object of interest is the minority class - namely the lesions [52]. Solutions for addressing this might be adjusting the loss function used in the algorithm or by creating a data set with a more balanced class distribution. The former can be accomplished by implementing cost-sensitive learning techniques to give additional weighting to correctly identifying lesion [53], however this is outside the scope of this investigation since it is not focused on effects of altering hyper parameters. The latter can be achieved by a technique known as oversampling [54]. This involves deconstructing the MRI volumes into slices and then cutting areas of slices with a more even distributing of lesion and non-lesion matter, preserving them for the dataset to be used in the training and testing which could be explored in detail in future work and refinement. An auxiliary benefit of this is that the time the algorithm would take to run would be reduced whilst simultaneously training an algorithm to more effectively recognise lesions. These points apply to the identification of smaller objects of interest across the medical field in the context of applying computational methods to medical object segmentation. The exclusion of both of the two smaller quartiles of lesions (lesions < 36 mm$^3$ in volume), demonstrated improved results which were statistically significant compared to the relevant control (Table 8). Dice results from Q3+Q4 for proton density images were, on average 0.5844 compared to the relevant control of 0.5726, a statistically significant improvement. The variation was not as large as anticipated. This could be because smaller lesions have less impact on the Dice score, as it correlates with the volume of pixels comprising a lesion. Conversely, if these lesions are removed from the ground truth of both training and testing data, there is also a likelihood that the FPR increased due to lesions which were identified by the creators of the ground truth not being included although this consideration was not followed up in this sub-study. Irregularity of lesion shape can impact segmentation in circumstances in which two lesions, one of which may be small, are connected by a relatively thin thread of demyelinated tissue and would therefore possibly be excluded from a particular sub-study. In some respects, this undermines the objective of the study and is clearly a limitation. In one related study, it was found that lesion volume can impact lesion detection by as much as 30% [28], whereas an 8% difference was recorded in this investigation. It should be noted, however, that in the referenced study, semi-automatic software was used as a basis for segmenting the ground truth. This was reviewed by two individuals with 30 years of experience in neurology and radiology, whereas the ground truth in the MICCAI 2016 dataset was annotated from scratch by seven different 'junior radiologists' with an average being taken to minimize bias [31]. The cited work was also conducted on a private dataset, making it difficult to demonstrate reliability by repetition because of issues of consent and confidentiality [28].

The heterogeneity in lesion presentations, due to differences in patients due to demographics, disease stage or duration, variation in subtype [10], can make lesion segmentation more challenging than more regularly presenting diseases such as tumours which also differ from MS in that they are masses of alien tissue as opposed to tissue scarring and thus easier to predict. The occurrence of different subtypes varies considerably with CIS, RRMS, SPMS disease course

comprising the majority of MS and the PPMS course only making up 10% to 15% [7]. This means that there is significantly less supply of PPMS data including MRI for both training algorithmic models and medical students. Machine learning techniques offer an avenue for supplementing data through generative methods, enabling the production of entirely original data.

## 5. Conclusion

In conclusion, this study comprehensively explores the impact of data adaptations in automatic MS lesion segmentation using the convolutional neural network, U-Net. Using multiple MRI modalities to train a model appeared to serve no benefit in improving segmentation results although it was apparent that a model trained on one image modality was capable of segmenting lesions in another. Increasing dataset size through rotational augmentation resulted in statistically significant improvement of results. Doing so more than once, however, resulted in no further improvement and in some instances even a decline illustrating the importance of maximising participation when creating datasets for studies such as this. Exclusion of lesions below or above certain volume thresholds influenced results directly. Excluding small lesions within the two lower quartiles led to incremental improvements. Whilst other medical segmentation tasks have, in essence, been solved; MS lesion segmentation remains challenging largely due to lack of consensus on ground truth, class imbalance, lesion size and irregularity and data availability. Addressing these issues could lead to the creation of a robust and dependable segmentation tool which could be used in a medical setting.

## Supporting information

**S1 Text. Contained in Supplementary Material 1 is a more detailed description of the neural network architecture used in this study on a layer by layer basis.**
(PDF)

**S2 Text. Contained in Supplementary Material 2 is additional data and experimental results.** Table A in S2 Text includes further results with the data redistributed between the training in the testing sets, breaking the rubric of the published challenge. Instead 38 or 43 brain volumes for training and 15 or 10 brain volumes for testing were used. Each imaging modality was looked at and each test was conducted 10 times. Table B in S2 Text contains data on the impact of removing the impact of a defective mask (patient 26) which contained no imaging mask in the set of ground truth data used. Table C in S2 Text contains results utilising different imaging modalities for both testing and training, examining every permutation (25 versions) each tested 10 times with an average taken.
(PDF)

## Acknowledgments

Generative AI played no role in the writing of this manuscript. The computations described in this paper were performed using the University of Birmingham's BlueBEAR HPC service, which provides a High Performance Computing service to the University's research community. The publicly available dataset used was collected by Observatoire Français de la Sclérose en Plaques (OFSEP), who was supported by a grant provided by the French State and handled by the "Agence Nationale de la Recherche", within the framework of the "Investments for the Future" program, under the reference ANR-10-COHO-002, by the Eugène Devic EDMUS Foundation against multiple sclerosis and by the ARSEP Foundation.

## Author contributions

**Conceptualization:** Adam Szekely, Luca Baronti, Daniel M Espino.

**Data curation:** Adam Szekely.

**Formal analysis:** Adam Szekely.

**Funding acquisition:** Adam Szekely, Marco Castellani, Daniel M Espino.

**Investigation:** Adam Szekely.

**Methodology:** Adam Szekely, Marco Castellani, Daniel M Espino.

**Resources:** Adam Szekely.

**Software:** Adam Szekely.

**Supervision:** Marco Castellani, Zubair Ahmed, Daniel M Espino.

**Visualization:** Adam Szekely.

**Writing – original draft:** Adam Szekely.

**Writing – review & editing:** Adam Szekely, Marco Castellani, Luca Baronti, Zubair Ahmed, William G K Manifold, Michael Douglas, Daniel M Espino.

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
