## [Decision Letter · Decision Letter 0]

23 Jun 2025

Response to Reviewers'. This file does not need to include responses to any formatting updates and technical items listed in the 'Journal Requirements' section below.'. This file does not need to include responses to any formatting updates and technical items listed in the 'Journal Requirements' section below.* A marked-up copy of your manuscript that highlights changes made to the original version. You should upload this as a separate file labeled 'Revised Manuscript with Track Changes'.'.* An unmarked version of your revised paper without tracked changes. You should upload this as a separate file labeled 'Manuscript'.'. If you would like to make changes to your financial disclosure, competing interests statement, or data availability statement, please make these updates within the submission form at the time of resubmission. Guidelines for resubmitting your figure files are available below the reviewer comments at the end of this letter. We look forward to receiving your revised manuscript. Kind regards, Avinash Singh, PhDAcademic EditorPLOS Digital Health Leo Anthony CeliEditor-in-ChiefPLOS Digital Healthorcid.org/0000-0001-6712-6626 **Journal Requirements:**

i. Please clarify all sources of funding (financial or material support) for your study. List the grants (with grant number) or organizations (with url) that supported your study, including funding received from your institution.

ii. State the initials, alongside each funding source, of each author to receive each grant.

iii. State what role the funders took in the study. If the funders had no role in your study, please state: “The funders had no role in study design, data collection and analysis, decision to publish, or preparation of the manuscript.”

iv. If any authors received a salary from any of your funders, please state which authors and which funders.

2. Please ensure that your Ethics Statement is available in its entirety at the beginning of your Methods section, under a subheading 'Ethics Statement'.

3. We ask that a manuscript source file is provided at Revision. Please upload your manuscript file as a .doc, .docx, .rtf or .tex.

4. Please upload separate figure files in .tif or .eps format. Also, remove the figures from your manuscript file but keep the legends.

5. We notice that your supplementary tables are included in the manuscript file. Please remove them and upload them with the file type 'Supporting Information'. Please ensure that each Supporting Information file has a legend listed in the manuscript after the references list.

6. Thank you for uploading your study's underlying data set. Unfortunately, the repository you have noted in your Data Availability statement does not qualify as an acceptable data repository according to PLOS's standards.

At this time, please upload the minimal data set necessary to replicate your study's findings to a stable, public repository (such as figshare or Dryad) and provide us with the relevant URLs, DOIs, or accession numbers that may be used to access these data. For a list of recommended repositories and additional information on PLOS standards for data deposition, please see

https://journals.plos.org/plosone/s/recommended-repositories.

7. Some material included in your submission may be copyrighted. According to PLOS’s copyright policy, authors who use figures or other material (e.g., graphics, clipart, maps) from another author or copyright holder must demonstrate or obtain permission to publish this material under the Creative Commons Attribution 4.0 International (CC BY 4.0) License used by PLOS journals. Please closely review the details of PLOS’s copyright requirements here: PLOS Licenses and Copyright. If you need to request permissions from a copyright holder, you may use PLOS's Copyright Content Permission form.

Potential Copyright Issues:

Figures 2, 4, 5, and 7: Please confirm whether you drew the images / clip-art within the figure panels by hand. If you did not draw the images, please provide (a) a link to the source of the images or icons and their license / terms of use; or (b) written permission from the copyright holder to publish the images or icons under our CC-BY 4.0 license. Alternatively, you may replace the images with open source alternatives. See these open source resources you may use to replace images / clip-art:

- https://openclipart.org/

**Additional Editor Comments (if provided):****Reviewers' Comments:** Reviewer's Responses to Questions

**Comments to the Author**

1. Does this manuscript meet PLOS Digital Health’s publication criteria? Is the manuscript technically sound, and do the data support the conclusions? The manuscript must describe methodologically and ethically rigorous research with conclusions that are appropriately drawn based on the data presented.? Is the manuscript technically sound, and do the data support the conclusions? The manuscript must describe methodologically and ethically rigorous research with conclusions that are appropriately drawn based on the data presented.

Reviewer #1: Yes

Reviewer #2: Yes

2. Has the statistical analysis been performed appropriately and rigorously?

Reviewer #1: N/A

Reviewer #2: Yes

3. Have the authors made all data underlying the findings in their manuscript fully available (please refer to the Data Availability Statement at the start of the manuscript PDF file)?

The PLOS Data policy requires authors to make all data underlying the findings described in their manuscript fully available without restriction, with rare exception. The data should be provided as part of the manuscript or its supporting information, or deposited to a public repository. For example, in addition to summary statistics, the data points behind means, medians and variance measures should be available. If there are restrictions on publicly sharing data—e.g. participant privacy or use of data from a third party—those must be specified.requires authors to make all data underlying the findings described in their manuscript fully available without restriction, with rare exception. The data should be provided as part of the manuscript or its supporting information, or deposited to a public repository. For example, in addition to summary statistics, the data points behind means, medians and variance measures should be available. If there are restrictions on publicly sharing data—e.g. participant privacy or use of data from a third party—those must be specified.

Reviewer #1: Yes

Reviewer #2: Yes

4. Is the manuscript presented in an intelligible fashion and written in standard English?

Reviewer #1: Yes

Reviewer #2: Yes

Reviewer #1: Title: Artificial Intelligence, Augmentation, Computational Methods, Dice, Machine Learning, Magnetic Resonance Imaging (MRI), Multiple Sclerosis (MS), Lesions

Restructure the abtract as shown below

Abtract

Background: Multiple Sclerosis (MS) is a demyelinating autoimmune disease of the central nervous system including the brain and spinal cord. Brain lesions are one of the most common radiological magnetic resonance imaging (MRI) features used both for diagnosis and clinical monitoring MS. Manually identifying and segmenting lesions is both difficult and time consuming, thus optimising approaches to reliable automatic segmentation is highly beneficial.

Objectives: The aim of this study was to assess the impact of data augmentation and manipulation on the accuracy of automated lesion segmentation using MRI scans from MS patients. Factors examined in this study include MRI modalities in both isolation and combination, image rotation, lesion size and size of testing set relative to training set.

Methods: Each factor was optimised and combined to maximise segmentation accuracy; the Dice metric was used as the focal metric to assess the efficacy of any given permutation of the setup. The statistical significance of results was assessed using the Mann-Whitney U-test and U-Net was chosen as the algorithmic method through which to segment lesions within the MRI volumes.

Results: The best Dice score achieved using the testing and training dataset as outlined in the MICCAI 2016 challenge rubric was 0.59, approximately a 2% improvement against controls. To achieve this result whilst adhering to the training and testing distribution as defined in the dataset publication, the optimal imaging sequence was determined to be proton density. The augmentation conditions used included implementing an additional rotation of the dataset (doubling it in size) and excluding lesions < 36.43 mm3 in volume.

Conclusions: A key finding of this study was that there was no statistically significant difference between using one MRI modality for training and another for testing. This suggests that a universal tool could be developed, trained on a single MRI type, and effectively applied in hospitals regardless of the specific MRI scans available to them, saving money and the time of clinicians.

Reviewer’s Comments

Background

In line 2 insert the word including after nervous system

Objectives

Kind state your objectives and leave out commentary

Methodology

Kindly summarize clearly the materials and methods used to carry out the study (Materials and methods)

Conclusion

Kindly link directely the objectives, methodology and results to the conclusion. What has been stated here is not the conclusion of this study

Introduction

Paragraph 1 Line 5 replace viewed with diagnosed

Paragraph 1 Line 6 replace “Their heterogeneity (shape, size, number, location) in the CNS” with “The heterogeneity (shape, size, number, location) of the CNS tissues”

Replace challenges with articles

Paragraph 2 line 4 insert “the” before analysis

Paragraph 2 line 8 replace “like” with “including”

Last paragraph line 4 insert “contrast medium” after enhancing

Was any material use?

Methods

Briefly explain the flowchart

Dataset

Kindly restructure the data collection procedure with each modality under the following

1. Image acquisition

2. Protocol use

3. procedure

Architecture

Provide step procedure of the architecture using figure 3

Experimental Procedure

Where are the link with the listed equations

Any application?

Results

Good presentation of the result

Discussions

link this statement with the results tables “The main findings of this study were (1) that initial testing, using the entire dataset, indicated that Flair was the optimal image modality and Gadolinium-enhanced images produced the lowest accuracy during the automated segmentation process of lesions”

which of the results sections are you discussing? Kindly be more specific

kindly discuss the results here not commentary

Impact of models

Provide more information on the potential impact on the application in medical imaging.

MS lesion segmentation specific Issues when using AI

Good discussions

Conclusion

Good conclusion

General comments

The study comprehensively explores the impact of data adaptations in automatic MS lesion segmentation using the convolutional neural network, U-net. Using multiple MRI modalities to train a model appeared to serve no benefit in improving segmentation results although it was apparent that a model trained on one image modality was capable of segmenting lesions in another. I think the study is a good one and fairly represent the study objectives.

However, these are some observations

1. The background to the introduction was too elaborate which include some reading materials available in literature I wonder what this adds to the study.

2. The methodology should include information on the materials used and brief explanation of the flowchart

3. The datset section should be restructured to be more systematic for easy understanding

4. The must be explanation of the applications of the various equations

5. The results were presented excellently

6. The discussions must be centred on the presented results not just commentary.

7. The conclusions were fairly done and answered the study objectives

Recommendation

The study can be accepted for publication if the suggested corrections are done.

References

Check your references. It is not good enough

Reviewer #2: This study employs the U-Net architecture to segment MRI data for Multiple Sclerosis (MS), which is an interesting and relevant topic. However, I have several concerns and suggestions that I hope the authors can address to improve the manuscript:

The authors chose U-Net for MS lesion segmentation, which is indeed a classic and widely used model. However, have the authors considered using nnU-Net, which is a self-adapting framework that has achieved state-of-the-art performance on various medical image segmentation tasks? A comparison or discussion would strengthen the methodology.

The dataset used—based on the 2016 MICCAI MS Lesion Segmentation Challenge—comprises data from only 53 MS patients, which seems rather limited for training a deep learning model. Are there any additional datasets available that could be used for external validation to enhance the robustness and generalizability of the results?

Have the authors employed 5-fold cross-validation or any other form of cross-validation during training to ensure the stability and reproducibility of the model’s performance?

Was test-time augmentation (TTA) used during inference to potentially improve segmentation accuracy? If not, the authors may consider incorporating it and reporting the impact.

Finally, the authors should further elaborate on the potential clinical implications of applying their AI model. Specifically, how could this approach benefit clinical practice, such as improving diagnostic efficiency, reducing radiologist workload, or supporting longitudinal monitoring of disease progression?

**Do you want your identity to be public for this peer review?** For information about this choice, including consent withdrawal, please see our Privacy Policy..

Reviewer #1: **Yes:** Dr Shirazu IssahakuDr Shirazu IssahakuDr Shirazu IssahakuDr Shirazu Issahaku

Reviewer #2: **Yes:** Ke ZhaoKe ZhaoKe ZhaoKe Zhao

**Figure resubmission:** While revising your submission, please upload your figure files to the Preflight Analysis and Conversion Engine (PACE) digital diagnostic tool, https://pacev2.apexcovantage.com/. PACE helps ensure that figures meet PLOS requirements. To use PACE, you must first register as a user. Registration is free. Then, login and navigate to the UPLOAD tab, where you will find detailed instructions on how to use the tool. If you encounter any issues or have any questions when using PACE, please email PLOS at figures@plos.org. Please note that Supporting Information files do not need this step. If there are other versions of figure files still present in your submission file inventory at resubmission, please replace them with the PACE-processed versions. **Reproducibility:** To enhance the reproducibility of your results, we recommend that authors of applicable studies deposit laboratory protocols in protocols.io, where a protocol can be assigned its own identifier (DOI) such that it can be cited independently in the future. Additionally, PLOS ONE offers an option to publish peer-reviewed clinical study protocols. Read more information on sharing protocols at https://plos.org/protocols?utm_medium=editorial-email&utm_source=authorletters&utm_campaign=protocols To enhance the reproducibility of your results, we recommend that authors of applicable studies deposit laboratory protocols in protocols.io, where a protocol can be assigned its own identifier (DOI) such that it can be cited independently in the future. Additionally, PLOS ONE offers an option to publish peer-reviewed clinical study protocols. Read more information on sharing protocols at https://plos.org/protocols?utm_medium=editorial-email&utm_source=authorletters&utm_campaign=protocols

---

## [Decision Letter · Decision Letter 1]

28 Jan 2026

Response to Reviewers'. This file does not need to include responses to any formatting updates and technical items listed in the 'Journal Requirements' section below.'. This file does not need to include responses to any formatting updates and technical items listed in the 'Journal Requirements' section below.* A marked-up copy of your manuscript that highlights changes made to the original version. You should upload this as a separate file labeled 'Revised Manuscript with Track Changes'.'.* An unmarked version of your revised paper without tracked changes. You should upload this as a separate file labeled 'Manuscript'.'. If you would like to make changes to your financial disclosure, competing interests statement, or data availability statement, please make these updates within the submission form at the time of resubmission. Guidelines for resubmitting your figure files are available below the reviewer comments at the end of this letter. We look forward to receiving your revised manuscript. Kind regards, Alexander WongSection EditorPLOS Digital Health Alexander WongSection EditorPLOS Digital Health Leo Anthony CeliEditor-in-ChiefPLOS Digital Healthorcid.org/0000-0001-6712-6626 **Journal Requirements:** If the reviewer comments include a recommendation to cite specific previously published works, please review and evaluate these publications to determine whether they are relevant and should be cited. There is no requirement to cite these works unless the editor has indicated otherwise.  **Additional Editor Comments (if provided):****Reviewers' Comments:** Reviewer's Responses to Questions

**Comments to the Author**

Reviewer #2: All comments have been addressed

Reviewer #3: (No Response)

publication criteria? Is the manuscript technically sound, and do the data support the conclusions? The manuscript must describe methodologically and ethically rigorous research with conclusions that are appropriately drawn based on the data presented.? Is the manuscript technically sound, and do the data support the conclusions? The manuscript must describe methodologically and ethically rigorous research with conclusions that are appropriately drawn based on the data presented.

Reviewer #2: Yes

Reviewer #3: No

3. Has the statistical analysis been performed appropriately and rigorously?

Reviewer #2: Yes

Reviewer #3: No

4. Have the authors made all data underlying the findings in their manuscript fully available (please refer to the Data Availability Statement at the start of the manuscript PDF file)?

The PLOS Data policy requires authors to make all data underlying the findings described in their manuscript fully available without restriction, with rare exception. The data should be provided as part of the manuscript or its supporting information, or deposited to a public repository. For example, in addition to summary statistics, the data points behind means, medians and variance measures should be available. If there are restrictions on publicly sharing data—e.g. participant privacy or use of data from a third party—those must be specified.requires authors to make all data underlying the findings described in their manuscript fully available without restriction, with rare exception. The data should be provided as part of the manuscript or its supporting information, or deposited to a public repository. For example, in addition to summary statistics, the data points behind means, medians and variance measures should be available. If there are restrictions on publicly sharing data—e.g. participant privacy or use of data from a third party—those must be specified.

Reviewer #2: Yes

Reviewer #3: Yes

5. Is the manuscript presented in an intelligible fashion and written in standard English?

Reviewer #2: Yes

Reviewer #3: Yes

Reviewer #2: The author has already addressed my concerns.

Reviewer #3: Thank you for your revision. However, the revision fails to adequately address several important concerns, particularly those highlighted by Reviewer #2. While the manuscript emphasizes data preprocessing and manipulation as its primary contribution, the current analysis does not convincingly demonstrate novelty or rigor, especially given the limited dataset and narrow scope of augmentation techniques. Below, I detail a few major concerns.

1. The author used a very basic U-Net model and did not provide a comparison with recent best-in-class models. Preprocessing effects cannot be interpreted meaningfully unless the underlying model is competitive and representative of current standards. When the author mentioned an improvement of 2% over MICCAI2016, did the author compare the model performance on the same footing, like excluding lesions < 36.43 mm³ in volume and well-stratified data in train, val, and test? Also, it would be helpful if the author can further clarify on threshold-deciding criteria for lesion size exclusion.

2. The data is limited, though it has multiple frame in one image; the author should produce 3-fold cross-fold validation or provide external validation set-based comparison for the current model and other state-of-the-art models. Re-running the same train/test split is not cross-validation and is unable to assess generalizability.

3. Combining multiple MRI modalities in training did not yield a clear benefit over the best single modality. The majority of multi-modal combinations produced no statistically significant improvement in Dice. The authors conclude that ’dataset hybridisation’ is not beneficial and the reason could be that the model had limited capacity or training data to exploit the extra information, or that some modalities introduced noise/redundancy. However, the author does not cross-check the results with other AI models or augmentations, which leads to contrasting findings and weakens the claims.

**Do you want your identity to be public for this peer review?** For information about this choice, including consent withdrawal, please see our Privacy Policy..

Reviewer #2: **Yes:** Ke ZhaoKe ZhaoKe ZhaoKe Zhao

Reviewer #3: No

**Figure resubmission:** While revising your submission, we strongly recommend that you use PLOS’s NAAS tool (https://ngplosjournals.pagemajik.ai/artanalysis) to test your figure files. NAAS can convert your figure files to the TIFF file type and meet basic requirements (such as print size, resolution), or provide you with a report on issues that do not meet our requirements and that NAAS cannot fix.

**Reproducibility:** To enhance the reproducibility of your results, we recommend that authors of applicable studies deposit laboratory protocols in protocols.io, where a protocol can be assigned its own identifier (DOI) such that it can be cited independently in the future. Additionally, PLOS ONE offers an option to publish peer-reviewed clinical study protocols. Read more information on sharing protocols at https://plos.org/protocols?utm_medium=editorial-email&utm_source=authorletters&utm_campaign=protocols To enhance the reproducibility of your results, we recommend that authors of applicable studies deposit laboratory protocols in protocols.io, where a protocol can be assigned its own identifier (DOI) such that it can be cited independently in the future. Additionally, PLOS ONE offers an option to publish peer-reviewed clinical study protocols. Read more information on sharing protocols at https://plos.org/protocols?utm_medium=editorial-email&utm_source=authorletters&utm_campaign=protocols

---

## [Editor Report · Decision Letter 2]

4 Mar 2026

Reliability of a convolutional neural network in segmenting Multiple Sclerosis lesions from MRI: Impact of data augmentation, image modality and tolerance with U-Net architecture

PDIG-D-25-00076R2

Dear Mr Szekely,

We are pleased to inform you that your manuscript 'Reliability of a convolutional neural network in segmenting Multiple Sclerosis lesions from MRI: Impact of data augmentation, image modality and tolerance with U-Net architecture' has been provisionally accepted for publication in PLOS Digital Health.

Best regards,

Dylan A Mordaunt, MD, MPH, FRACP

Section Editor

PLOS Digital Health

**Additional Editor Comments (if provided):**

Thank you for your revised manuscript, "Reliability of a convolutional neural network in segmenting Multiple Sclerosis lesions from MRI: Impact of data augmentation, image modality and tolerance with U-Net architecture" (PDIG-D-25-00076R2).

I have reviewed your rebuttal along with the feedback from Reviewer 3. Following this assessment, I am pleased to recommend accepting your manuscript for publication in PLOS Digital Health.

Below are some final editorial remarks regarding the exchange with Reviewer 3:

Editorial Assessment of Reviewer 3's Comments:

We acknowledge Reviewer 3’s valid points raised from a pure machine-learning evaluation standpoint- specifically the calls for cross-fold validation and benchmarking against state-of-the-art architectures (like nnU-Net). In pure computer science venues, relying on a single static train/test split (especially given the total sample size of 53 patients, despite the high slice count) often limits claims of broad generalizability.

However, from an applied digital health and experimental design perspective, your argument holds weight. The core contribution of your work is an ablative study on data preprocessing (augmentation, modality, and lesion size) using a fixed, established baseline model (U-Net). Introducing newer, complex architectures could indeed introduce confounding variables that mask the direct clinical impact of the data manipulations you are studying. We appreciate that you have acknowledged these methodological limitations in your discussion and have appropriately directed future studies toward exploring these data manipulations on newer architectures and with expanded validation strategies.

Because the manuscript clearly articulates its scope and its clinical applications, we find your rationale sufficient for this journal.